# Hierarchical Self-Attention: Generalizing Neural Attention Mechanics to Multi-Scale Problems

**Saeed Amizadeh, Sara Abdali, Yinheng Li, and Kazuhito Koishida**
Microsoft
Redmond, WA 98052
{saamizad, saraabdali, yinhengli, kazukoi}@microsoft.com

## Abstract

Transformers and their attention mechanism have been revolutionary in the field of Machine Learning. While originally proposed for the language data, they quickly found their way to the image, video, graph, etc. data modalities with various signal geometries. Despite this versatility, generalizing the attention mechanism to scenarios where data is presented at different scales from potentially different modalities is not straightforward. The attempts to incorporate hierarchy and multi-modality within transformers are largely based on ad hoc heuristics, which are not seamlessly generalizable to similar problems with potentially different structures. To address this problem, in this paper, we take a fundamentally different approach: we first propose a mathematical construct to represent multi-modal, multi-scale data. We then mathematically *derive* the neural attention mechanics for the proposed construct from the first principle of *entropy minimization*. We show that the derived formulation is *optimal* in the sense of being the closest to the standard Softmax attention while incorporating the inductive biases originating from the hierarchical/geometric information of the problem. We further propose an efficient algorithm based on dynamic programming to compute our derived attention mechanism. By incorporating it within transformers, we show that the proposed hierarchical attention mechanism not only can be employed to train transformer models in hierarchical/multi-modal settings from scratch, but it can also be used to inject hierarchical information into classical, pre-trained transformer models *post training*, resulting in more efficient models in zero-shot manner.

## 1 Introduction

The field of Deep Learning has recently experienced a spectacular breakthrough with the rise of Large Language Models (LLMs). It is no secret that this success is largely owed to the Transformer architecture [90] and its self-attention mechanism. Although they were originally proposed to work with language [24, 52, 13, 18], transformers have found their way to deal with images [25, 100, 85], video [9, 65, 14, 46], audio [32, 91, 45, 16], graphs [101, 61, 72, 76], groups [42, 83], manifolds [38] and point clouds [33, 107] *without* significantly altering their basic neural attention mechanism. This is mainly due to the fact that, unlike many other architectures, transformers incorporate data geometry not by architectural priors but by explicit, black-box, position embedding functions, which can be easily replaced from one domain to another.

Despite this versatility, information quite often comes in different modalities and at different scales. In terms of geometry, this means that we deal with problems where each datapoint may occupy multiple, mutually-inconsistent geometries at potentially different scales. This is indeed challenging, even for transformers! To this end, various novel (often heuristic) neural architectures have been proposed to deal with multi-modal [55, 44, 109, 87, 23, 103, 40, 69] and hierarchical data [54, 105, 67, 94, 19, 106, 110].

39th Conference on Neural Information Processing Systems (NeurIPS 2025).

Aside from the heuristic-based nature of many of these empirical architectures, they quite often suffer from a more practical dilemma. On one hand, many such frameworks tend to partially discard geometrical or hierarchical information depriving the learning task from valuable domain knowledge which can significantly reduce the model's statistical complexity. On the other hand, by incorporating the full geometrical knowledge of different modalities and their hierarchical structure within these heuristic frameworks, we often end up with highly problem-specific architectures that are hardly generalizable to other similar problems. To address this challenge in a unified and principled way, in this work, we take a radically different approach:

- Instead of coming up with yet another heuristic neural architecture right off the bat, we first propose a mathematical construct called *nested signal* to formally represent multi-geometry, hierarchical information. As we show, the proposed formalism enables us to coherently represent different geometrical domains at different scales while maintaining its generality across different problems.

- In order to define mathematically-sound neural operations on nested signals, we turn to the attention mechanism. First, we show that the standard Softmax self-attention [90] can be mathematically derived from the principle of *entropy minimization*. Then by generalizing this principle to nested signals, we derive the *hierarchical self-attention (HSA)* neural mechanics which is the generalization of the Softmax attention for nested signals.

- We further show that the attention weights derived from the HSA are *optimal* in the sense of being the closest to flat Softmax attention weights in terms the total KL-divergence, while at the same time adhering to the hierarchical structure of the data.

- Next, we propose an efficient algorithm based on *dynamic programming* to calculate the HSA, that is provably faster than its direct evaluation. By implementing HSA within the transformer architecture, we empirically show that we are able to train models that can seamlessly incorporate the hierarchical/multi-modal domain knowledge to arrive at better and more efficient transformers.

- Last but not least, we show that HSA can further replace the standard Softmax self-attention operation in pre-trained transformers and significantly reduce the number of self-attention FLOPs while incurring minimal Accuracy drop, in an entirely *zero-shot* manner.

## 2 Representing Hierarchical Data

In Geometric Deep Learning, a *signal* $x$ is defined as the mapping $x : \Omega \to \mathcal{C}$, where the set $\Omega$ is the *domain* of the signal and $\mathcal{C}$ is a vector space, typically $\mathbb{R}^d$ with $d$ being the *channel* dimension. For example, an RGB image is a signal where $\Omega$ is the 2D grid and $\mathcal{C} = \mathbb{R}^3$, *i.e.* the RGB color space. Similarly, text can be seen as a signal with $\Omega$ being the 1D grid and $\mathcal{C}$ a word embedding space. More niche applications in Geometric Deep Learning [17] extend the notion of signals to the domain of graphs, gauges, manifolds, etc. by defining the appropriate structure for $\Omega$. We refer to the set of all such possible domains as $\mathcal{D}$. The elements $\Omega \in \mathcal{D}$ are not necessarily vector spaces (*e.g.* 2D grid). In order to numerically handle these spaces, we define a special signal $\varepsilon_\Omega : \Omega \to \mathbb{R}^c$ for each $\Omega \in \mathcal{D}$ which maps the elements of each domain in $\mathcal{D}$ to $\mathbb{R}^c$; we refer to this special signal as the *position embedding*. Given $\varepsilon_\Omega$, each signal $x$ defined on $\Omega$ is seen as $x : \varepsilon_\Omega(\Omega) \to \mathcal{C}$. In Appendix C, we generalize the notion of signal to encompass traditional tabular features.

In this section, we introduce the notion of *nested signals* which is the key modeling tool to represent multi-modal, hierarchical data. To this end, we first define the set of all *simple* signals $\mathcal{S}$ as the set of all possible signals defined on all possible domains; that is, $\mathcal{S} = \{x : \Omega \to \mathcal{C} \mid \Omega \in \mathcal{D}\}$. The signals defined on different domains may have different channel dimensions; to make the channel dimension uniform across different domains, we zero-pad the lower dimensional signals to the maximum channel dimensionality $d$ across different domains, such that each element of $\mathcal{S}$ has the same channel dimension $d$ regardless of its domain.

**Definition 2.1** (Nested Signal)**.** The set of $d$-dimensional nested signals up to depth $\ell$, $\mathcal{N}_\ell$, is recursively defined as $\mathcal{N}_\ell = \{x : \Omega \to \mathcal{U} \mid \Omega \in \mathcal{D}, \mathcal{U} \in \{\mathcal{N}_{\ell-1}, \mathbb{R}^d\}\}$, where $\mathcal{N}_0 = \mathbb{R}^d$. Also, define $\mathcal{N} = \mathcal{N}_\ell$ as $\ell \to \infty$; each element $x \in \mathcal{N}$ is then referred to as a *nested signal*. The top-level domain $\Omega \in \mathcal{D}$ of a nested signal $x$, denoted by $r(x)$.

For example, a website is a nested signal where at the top level, we have webpages defined on the nodes of a graph domain representing the link structure between the webpages. Each webpage is in turn another nested signal where at its top level we have

an unordered set of textboxes and images constituting the page. Going one level further, each textbox or image is a (simple) signal assigning word embeddings or pixel values to the nodes of 1D or 2D grid domains, respectively. Fig. 1(Top) depicts this example.

While in theory, the domains $\Omega \in \mathcal{D}$ can be infinite, in practice, we mostly deal with nested and simple signals defined on finite $\Omega$'s. In particular, a nested signal $x$ is said to be *finite* if the domains $\Omega$'s at *all* of its nesting levels are finite. Given the set of position embeddings $\varepsilon = \{\varepsilon_\Omega \mid \Omega \in \mathcal{D}\}$, a finite nested signal can be represented by a *signal hierarchy* as follows:

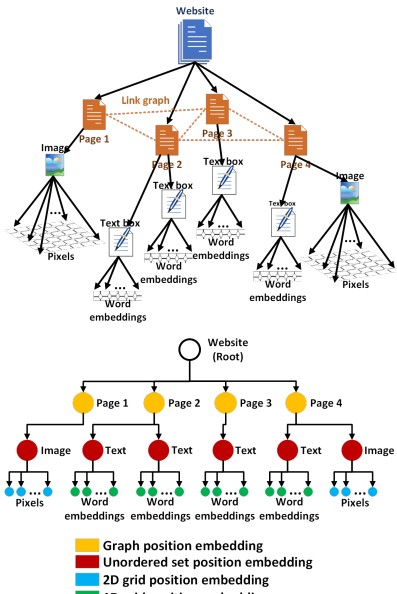

**Definition 2.2** (Signal Hierarchy). For a finite nested signal $x$, its signal hierarchy $h_x$ is a tree with the root node $R_x$ associated with $r(x)$, the top level domain of $x$. The children of $R_x$ are defined as $chd(R_x) = \{h_{x(u)} \mid u \in r(x)\}$ where $x(u)$ is the value of signal (possibly another nested signal) at $u$. If $x(u)$ is a vector instead of a signal, then $h_{x(u)}$ is simply $x(u)$. Furthermore, each child $h_{x(u)} \in chd(R_x)$ is annotated by $\varepsilon_{r(x)}(u)$, the position embedding vector dictated by its parent node.

We denote the nodes (and equivalently their corresponding sub-trees) in the signal hierarchy $h_x$ by upper-case letters. Any set of sibling nodes in $h_x$ is referred as a *family*. The members of a family are nested signals (or real vectors for the leaf nodes) that reside on the same domain $\Omega$ and therefore share the same position embedding function $\varepsilon_\Omega$. Furthermore, for $A \in h_x$, $chd(A)$, $sib(A)$ and $\ell(A)$ represent the set of $A$'s children, its siblings and the index set of the leaf node descendants of $A$, respectively. Two nodes in $h_x$ are called *unrelated* if neither of them is descendant of the other. For two unrelated nodes $A$ and $B$, their *immediate common ancestor* is denoted by

Figure 1: **(Top)** A nested signal representing a website. **(Bottom)** Its signal hierarchy representation. Different colors encode different types of position embeddings assigned to each node.

$ica(A, B)$, while their *highest distinct ancestors* are denoted by $A'$ and $B'$, respectively, where we have $A', B' \in chd(ica(A, B))$; *i.e.*, $A'$ and $B'$ are always siblings even if $A$ and $B$ are not. See Appendix A for the notational details as well as a visual demonstration of the tree-related concepts.

Since sibling nodes share the same position embedding function, the relative positional distance (or similarity) between them is well-defined. More generally, for any two unrelated nodes $A, B \in h_x$, we can form a well-defined positional distance between them by comparing the position embeddings of $A'$ and $B'$ which is well-defined since $A'$ and $B'$ are always siblings. The implication of this construction is indeed powerful as it would enable the signal hierarchy formalism to define meaningful positional distance between any two unrelated nodes in the hierarchy *regardless of their modalities or signal types*. Fig. 1(Bottom) shows the signal hierarchy representation for our earlier website example.

## 3 Hierarchical Self-Attention

The nested signal formalism and its signal hierarchy representation introduced in the previous section provide a systematic way to represent hierarchical data that can potentially span across different modalities and domains. However, the question remains what kind of neural architectures can handle such versatile data structure? To answer this question, we note that for non-hierarchical, simple signals, the transformer architecture first introduced by [90] allows for a unified representation learning methodology that can accommodate various signal domains (as long as the position embedding is available), not to mention its remarkable success in revolutionizing deep learning. However, extending the attention mechanism to nested signals is not straightforward as the information in such signals can come with different signal domains at different scales.

To address this problem, in this section, we first propose a statistical mechanical framework that elegantly derives the classical Softmax attention mechanism from the principle of entropy minimization when a finite (simple) signal is viewed as a physical system with $N$ particles. By generalizing our proposed construction to nested systems, we then derive a novel, theoretically-rigorous mechanism

for calculating self-attention within nested signals, which we refer as *Hierarchical Self-Attention (HSA)*. By its direct construction, the proposed HSA mechanism aims at reducing the total entropy of the nested system, or equivalently put, *increasing information within the learned representation of the nested signal*. We further show that our proposed construction to derive HSA is *optimal* in the sense of Kullback-Leibler (KL) divergence from the Softmax attention weights if the hierarchical structure were to be ignored. This result will subsequently open the door for the application of our proposed formulation to *approximate* the inefficient Softmax attention in pre-trained transformers using the more efficient hierarchical calculations if a hierarchy exists and can be imposed in a given problem. Finally, we propose an efficient algorithm based on dynamic programming that calculates HSA for a given signal hierarchy $h_x$ in $O(M \cdot b^2)$, where $M$ is the number of families in $h_x$ and $b$ is its maximum branching factor (*i.e.* family size).

## 3.1 Softmax Attention Revisited

Let $x = \{x_i \in \mathbb{R}^{d'} \mid 1 \leq i \leq N\}$ be a finite signal with $N$ elements in $\mathbb{R}^{d'}$ with the corresponding position embeddings $\{e_i \in \mathbb{R}^c \mid 1 \leq i \leq N\}$. To calculate self-attention over $x$, one needs to define the set of *query* variables $Q = \{q_i \in \mathbb{R}^d \mid 1 \leq i \leq N\}$ and *key* variables $K = \{k_i \in \mathbb{R}^d \mid 1 \leq i \leq N\}$, where $q_i$'s and $k_i$'s are (linear) functions of $x_i$. Then the conditional entropy of $Q$ given $K$ is:

$$\mathrm{H}(Q \mid K) = -\int \wp(Q, K) \log \wp(Q \mid K) dQ dK = -\mathbb{E}_{Q,K}\big[\log \wp(Q \mid K)\big] \tag{1}$$

where $\wp(Q, K)$ and $\wp(Q \mid K)$ are the unknown joint and posterior distributions over $Q$ and $K$. While the joint distribution can be approximated using the Monte Carlo method, the posterior can be approximated by a variational distribution $\xi(Q \mid K)$, which gives rise to the variational upper-bound on the conditional entropy:

$$\mathrm{H}_{UB}(Q \mid K) = -\mathbb{E}_{Q,K}\big[\log \xi(Q \mid K)\big] \geq \mathrm{H}(Q \mid K) \tag{2}$$

We further represent the variational distribution by the Boltzmann distribution, *i.e.* $\xi(Q \mid K) = \frac{1}{Z(K)} \exp[-\phi(Q, K)/\tau]$, where $\phi(Q, K)$, $Z(K)$, and $\tau$ are the energy function[1], the partition function and the temperature parameter, respectively. The variational upper-bound then can be written as:

$$\mathrm{H}_{UB}(Q \mid K) = \mathbb{E}_{Q,K}\big[\phi(Q, K)/\tau\big] + \mathbb{E}_K\big[\log Z(K)\big] \tag{3}$$

The end goal of representation learning is to transform the input signal (*i.e.* the query variables $Q$) into a "better" representation. A principled way to arrive at a better representation is to modify $Q$ such that its information content is maximized, or equivalently its entropy is minimized. Since we cannot directly calculate the entropy, we can work with its variational upper-bound $\mathrm{H}_{UB}$ as a proxy. Then, the entropy minimization approach amounts to gradient descent on $\mathrm{H}_{UB}$ w.r.t. each $q_i$:

$$q_i \leftarrow q_i - \lambda \cdot \nabla_{q_i}\mathrm{H}_{UB}(Q \mid K) = q_i - \lambda \cdot \mathbb{E}_{Q,K}\left[\frac{1}{\tau}\nabla_{q_i}\phi(Q, K)\right], \quad 1 \leq i \leq N \tag{4}$$

where $\lambda > 0$ is the step size.

**Proposition 3.1 (Softmax Attention).** *For the energy function $\phi(Q, K)$, defined as:*

$$\phi(Q, K) = -\frac{1}{N}\sum_{i=1}^{N}\log\left(\frac{1}{N-1}\sum_{j=1, j\neq i}^{N}\exp\Big[\frac{-1}{2\sqrt{d}}\|q_i - k_j\|^2 + e_i^T e_j\Big]\right) \tag{5}$$

*if both Q and K variables are normalized using the LayerNorm function [10], then for $\tau = (N\sqrt{d})^{-1}$, $\lambda = 1$ and sample size of $1$, the Eq. (4) reduces to:*

$$1 \leq i \leq N \,,\, q_i \leftarrow q_i + \sum_{j=1, j\neq i}^{N}\frac{\exp(q_i^T k_j/\sqrt{d} + e_i^T e_j)}{\sum_{t=1, t\neq i}^{N}\exp(q_i^T k_t/\sqrt{d} + e_i^T e_j)} \cdot k_j \tag{6}$$

*essentially, this is Softmax attention via residual connection.*

*Proof.* See Appendix H.1. □

---
[1]Note that the energy function needs to satisfy $\int \exp[-\phi(Q, K)/\tau]dQ < \infty$.

Note that (6) is similar to the original attention formulation proposed by [90], except for a few differences: (1) there is no separate *value* linear projection; the value projection emerges later as we incorporate learnable step-size (see Appendix D), (2) the LayerNorm is applied post-linear projection as opposed to pre-normalization in the original formulation, and (3) the residual addition is applied post-linear projection. In other words, with few minor modifications, the original Softmax attention operation can be interpreted as maximizing the information content in the representation. But the real importance of the formulation in (4) is that depending on how we define the energy function, we can arrive at various types of attention mechanisms tailored to different applications. We use this feature in the next section to derive a hierarchical self-attention (HSA) mechanism for nested signals.

## 3.2 Generalizing Attention to Nested Signals

We derive a self-attention mechanism for finite nested signals represented via a signal hierarchy tree. We follow the same recipe as the previous section by defining an appropriate energy function. But first, for any two unrelated nodes $A$ and $B$ in $h_x$, we define the *interaction energy* $\psi_{A \to B}$:

$$\psi_{A \to B} = -\varepsilon_\Omega(A')^T \varepsilon_\Omega(B') + \frac{1}{2\sqrt{d} \cdot |\ell(A)| \cdot |\ell(B)|} \sum_{i \in \ell(A)} \sum_{j \in \ell(B)} \|q_i - k_j\|^2 \tag{7}$$

where $|\cdot|$ denotes set cardinality, $\varepsilon_\Omega(\cdot)$ is the position embedding dictated by $ica(A,B)$ (*i.e.* $\Omega = r(ica(A,B))$), and $A'$ and $B'$ are the highest distinct ancestors of $A$ and $B$, as defined in Section 2. Intuitively speaking, the interaction energy $\psi_{A \to B}$ captures the *dissimilarity* between the nested signals rooted at $A$ and $B$ as a weighted sum of their highest non-common ancestors' position dissimilarity (the first term) and the average Euclidean distance between their leaf nodes (the second term). By calculating energy (dissimilarity) at the subtree level instead of individual leaves, we inherently encode the inductive bias that the leaf nodes of a subtree (*i.e.* a nested signal) can be *pooled* into a single representative (*i.e.* the subtree's root) while roughly maintaining the underlying semantics. This is referred to as *scale separation* in Geometric Deep Learning [17], a fundamental prior in dealing with multi-scale physical systems, benefiting us both statistically (by taming the curse of dimensionality) and computationally (by providing efficient algorithms). Using the interaction energy definition, the energy of the signal hierarchy rooted at non-leaf node $A$ is *recursively* defined:

$$\phi(A) = - \sum_{B \in chd(A)} \frac{|\ell(B)|}{|\ell(A)|} \log \left[ \exp\left(-\phi(B)\right) + \sum_{C \in sib(B)} |\ell(C)| \exp\left(-\psi_{B \to C}\right) \right] \tag{8}$$

For leaf nodes, $\phi(A)$ is set to $\infty$. $\phi(R_x)$ is the energy of the whole signal hierarchy $h_x$. Intuitively, (8) states that the energy of a system (a signal hierarchy tree) is the weighted sum of the energy contribution of its subsystems (immediate subtrees) where the weights are proportional to the size of each subsystem. The contribution of each subsystem, in turn, is a non-linear combination (via the *weighted* **log-sum-exp** function, which is the addition operation in the log-space) of the energy of the subsystem itself (the recursion term) and its interactions with its sibling subsystems (the second term). It is easy to see that for single-level $h_x$ (*i.e.* simple signals), $\phi(R_x)$ reduces to (5). Having defined the energy function, we can follow the recipe in (4) to calculate the HSA for $h_x$ by recursively computing the gradients $\nabla_{q_i} \phi(R_x) \in \mathbb{R}^d$ for each leaf node $q_i$, $i \in \ell(R_x)$ as:

$$\nabla_{q_i} \phi(R_x) = \left[ \frac{\alpha(B^i) \cdot \nabla_{q_i} \phi(B^i) + \sum_{C \in sib(B^i)} |\ell(C)| \beta(B^i, C) \cdot \nabla_{q_i} \psi_{B^i \to C}}{\alpha(B^i) + \sum_{C \in sib(B^i)} |\ell(C)| \beta(B^i, C)} \right] \cdot \frac{|\ell(B^i)|}{|\ell(R_x)|} \tag{9}$$

$$\text{where } \alpha(B^i) = \exp\left(-\phi(B^i)\right), \beta(B^i, C) = \exp\left(-\psi_{B^i \to C}\right) \tag{10}$$

and $B^i$ denotes the child of $R_x$ which contains $q_i$ as a leaf. It is not difficult to show that for the quadratic interaction energy function in (7), if both $Q$ and $K$ variables are normalized beforehand using a LayerNorm layer, then the recurrence in (9) can be *unrolled* and written in the matrix form (see (36) in Appendix H.3) $\nabla\Phi = \Theta K$, where :

$$\nabla\Phi = [\nabla_{q_1} \phi(R_x), ..., \nabla_{q_{|\ell(R_x)|}} \phi(R_x)]^T, \quad K = [k_1, ..., k_{|\ell(R_x)|}]^T \tag{11}$$

and $\Theta = [\theta_{i,j}]_{|\ell(R_x)| \times |\ell(R_x)|}$ is the *attention matrix*; that is, $\theta_{i,j}$ is the coefficient of the key variable $k_j$ for computing the attention update $\nabla_{q_i} \phi(R_x)$ for the query variable $q_i$ in (9). However, $\Theta$ is different from classical attention matrix in the sense that many of its entries share the same values. In

particular, for any two sibling nodes $A$ and $B$ in $h_x$, the corresponding entries between the leaves of $A$ and $B$ form a *block* in $\boldsymbol{\Theta}$ with one value; that is, $\theta_{i,j} = \theta_{A,B}, \forall i \in \ell(A), j \in \ell(B)$. In other words, the attention weight between any leaf node in $A$ and any leaf node in $B$ is approximated by one value $\theta_{AB}$; we refer to this approximation between the leaves of sibling nodes in $h_x$ as the *block constraint* which makes the attention matrix a *hierarchical matrix* [35, 36]. Fig. 2(Left) illustrates the self-attention matrix for a toy example signal hierarchy with the block constraint. The block constraint is directly administered by the form of the interaction energy function in (7) as well as the signal hierarchy energy recurrence in (8).

The block constraint effectively reduces the degrees of freedom for an attention matrix from $O(|\ell(R_x)|^2) = O(M^2 \cdot b^2)$ to $O(M \cdot b^2)$, where $|\ell(R_x)|$, $M$ and $b$ are the total number of leaf nodes, the number families (*i.e.* non-leaf nodes) and the maximum branching factor in $h_x$, respectively. Without it, we essentially go back to the standard Softmax attention mechanism where the unnormalized attention weights before Softmax are calculated by evaluating the interaction energy function for every pair of leaf nodes. We refer to this process as *flattening* a nested signal. Fig 2(Right) shows the self-attention matrix for the flattened version of our earlier toy example *without* the block constraint. Flattening is not only computationally costly (by being quadratic in $M$ instead of linear), it may also hurt the model statistically. Note that by enforcing coarse-grained attention weights through the block constraint, we effectively administer a form of regularization guided by the scale separation prior which is in turn induced from the prior knowledge of the hierarchical structure in the problem. By flattening a nested signal, we simply discard this prior knowledge which can make the model prone to over-fitting.

Note that the block constraint by itself merely enforces tied values for the attention weights over the leaves of sibling nodes; it does *not*, however, specify what those values should be. That is, there are infinitely many attention matrices that adhere to the block constraint; our proposed formulation in (9) is just one of them. However, as we show next, our proposed formulation is *optimal* in the sense of being the closest approximation to the standard Softmax attention if the nested signal were to be treated as a flat, simple signal.

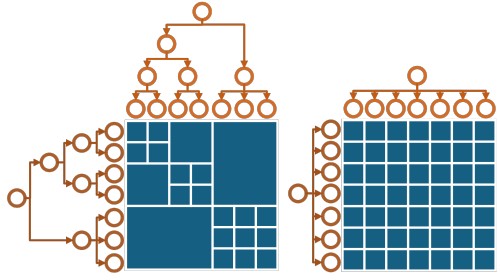

Figure 2: **(Left)** The self-attention matrix for a toy signal hierarchy with the block constraint. Each contiguous tile here represents one tied value for the corresponding cells. **(Right)** The self-attention matrix for the flattened (or simple) signal without the block constraint.

**Theorem 3.2** (**The optimality of HSA**). *Let both $Q$ and $K$ variables be normalized using the LayerNorm function. For the given interaction energy function $\psi$ in* (7)*, if $\boldsymbol{\Theta} = [\theta_{i,j}]_{|\ell(R_x)| \times |\ell(R_x)|}$ is the self-attention matrix for the nested signal $x$ derived from the proposed gradient recurrence in* (9) *(as depicted by* (11)*), then for the temperature parameter $\tau = \left( |\ell(R_x)| \sqrt{d} \right)^{-1}$, $\hat{\boldsymbol{\Theta}} = -\frac{1}{\tau}\boldsymbol{\Theta}$ is a stochastic matrix; that is, it is non-negative and we have $\hat{\boldsymbol{\Theta}}\mathbf{1} = \mathbf{1}$. Moreover, $\hat{\boldsymbol{\Theta}}$ is the closest attention matrix with the block constraint to the classical Softmax attention matrix for the flattened signal in terms of total KL-divergence; that is,*

$$\hat{\boldsymbol{\Theta}} = \arg\min_{\boldsymbol{\Theta} \in \mathcal{B}} \sum_{i \in \ell(R_x)} D_{KL}(\theta_{i,\cdot} \| \theta_{i,\cdot}^f) \tag{12}$$

*where $\mathcal{B} \subset \mathbb{R}^{|\ell(R_x)| \times |\ell(R_x)|}$ is the space of all stochastic attention matrices that admit the block constraint induced by $h_x$, and $\theta_{i,\cdot}^f$ ($\forall i \in \ell(R_x)$) are the rows of the attention matrix for the flattened version of the signal:*

$$\theta_{i,j}^f = \frac{\exp(-\psi_{i \to j})}{\sum_{k \in \ell(R_x), k \neq i} \exp(-\psi_{i \to k})} , \ \forall i, j \in \ell(R_x) \tag{13}$$

*Proof.* See Appendix H.2. □

This result is crucial in the sense that it shows our proposed HSA mechanism for nested signals formalized by (8) and (9) is the *closest approximation* to the classical attention mechanism while at the same time adhering to the block constraint (induced by the hierarchical structure of the nested signal), which benefits the model computationally *and* statistically.

From the practical perspective, this result has another important implication: if we replace the interaction energy function $\psi_{i \to j}$ with the original cosine similarity in transformers (where the

position information is simply *added* to the signal), our proposed methodology provides the *closest* hierarchical approximation of the original Softmax attention. Practically speaking, this means that if we have access to some form of hierarchical information $h_x$ in a problem at inference time, we can simply replace the self-attention operation in pre-trained transformer-based models by HSA and arrive at much more efficient calculations *without the need for major re-training*. Note that the direct evaluation of the recurrence in (9) for all query variables $q_i$ still takes $O(b^2 \cdot M \log_b M)$. In Appendix E, we prove that the HSA can be computed in $O(M \cdot b^2)$ using a dynamic programming algorithm. Furthermore, we propose a transformer encoder architecture based on the HSA in Appendix F.

# 4 Efficient Calculation of HSA

---

**Algorithm 1:** Hierarchical Self-Attention

1 **Input:** $h_x$
2 **Output:**
  $\{\nabla_{q_i}\phi(R_x) \in \mathbb{R}^d, \forall i \in \ell(R_x)\}$
3 $u \leftarrow -\log(|\ell(R_x)|)$
4 `ComputeSufficientStats`$(R_x)$
5 `ComputeAttention`$(R_x, u, \mathbf{0})$
6 **foreach** $i \in \ell(R_x)$ **do**
7 $\quad \nabla_{q_i}\phi(R_x) \leftarrow \vartheta(L_i)$
8 **end**
9 **return** $\{\nabla_{q_i}\phi(R_x) \mid i \in \ell(R_x)\}$

---

**Algorithm 2:** The Top-down Pass

1 **Input:** $A \in h_x, u \in \mathbb{R}, \mathbf{v} \in \mathbb{R}^d$
2 **Output:** $\vartheta(A) \in \mathbb{R}^d$
3 **Function** `ComputeAttention`$(A, u, \mathbf{v})$:
4 $\quad$ **foreach** $C \in chd(A)$ **do**
5 $\quad\quad \vartheta(C) \leftarrow \mathbf{v} - \frac{1}{\sqrt{d}} \exp\big(u +$
  $\quad\quad \text{LogSigmoid}\big[\phi(C) - \eta(C)\big]\big) \cdot \vartheta(C)$
6 $\quad\quad u' \leftarrow u + \text{LogSigmoid}\big[\eta(C) - \phi(C)\big]$
7 $\quad\quad$ `ComputeAttention`$(C, u', \vartheta(C))$
8 $\quad$ **end**
9 **End Function**

---

**Algorithm 3:** The Bottom-up Pass

1 **Input:** $A \in h_x$                 `// A node in the signal hierarchy`
2 **Output:** $\phi(A) \in \mathbb{R}, \eta(A) \in \mathbb{R}, \vartheta(A) \in \mathbb{R}^d$
3 **Function** `ComputeSufficientStats`$(A)$:
4 $\quad$ **if** *A is a leaf* **then**
5 $\quad\quad \phi(A) \leftarrow \infty$
6 $\quad\quad \rho_q(A) \leftarrow q(A)$                      `// q(A) is the query at leaf A`
7 $\quad\quad \rho_k(A) \leftarrow k(A)$                      `// k(A) is the key at leaf A`
8 $\quad\quad \rho_v(A) \leftarrow v(A)$                      `// v(A) is the value at leaf A`
9 $\quad$ **else**
10 $\quad\quad$ **foreach** $C \in chd(A)$ **do**
11 $\quad\quad\quad$ `ComputeSufficientStats`$(C)$
12 $\quad\quad$ **end**
13 $\quad\quad \phi(A) \leftarrow -\sum_{C \in chd(A)} \frac{|\ell(C)|}{|\ell(A)|} \cdot \log\big[\exp\big(-\phi(C)\big) + \exp\big(-\eta(C)\big)\big]$
14 $\quad\quad \rho_q(A) \leftarrow \frac{1}{|\ell(A)|} \sum_{C \in chd(A)} |\ell(C)|\rho_q(C)$
15 $\quad\quad \rho_k(A) \leftarrow \frac{1}{|\ell(A)|} \sum_{C \in chd(A)} |\ell(C)|\rho_k(C)$
16 $\quad\quad \rho_v(A) \leftarrow \frac{1}{|\ell(A)|} \sum_{C \in chd(A)} |\ell(C)|\rho_v(C)$
17 $\quad$ **end**
18 $\quad \forall B \in sib(A) : \psi'_{A \to B} \leftarrow \varepsilon(A)^T \varepsilon(B) + \frac{1}{\sqrt{d}}\rho_q(A)^T \rho_k(B) - \sqrt{d} + \log|\ell(B)|$
19 $\quad \eta(A) \leftarrow -\log\big[\sum_{B \in sib(A)} \exp(\psi'_{A \to B})\big]$
20 $\quad \vartheta(A) \leftarrow \exp\big(-\eta(A)\big) \sum_{B \in sib(A)} \exp(\psi'_{A \to B}) \cdot \rho_v(B)$
21 **End Function**

---

Even though our proposed HSA formulation in Eq. (9) brings down the degrees of freedom for the attention matrix to $O(M \cdot b^2)$, the naïve implementation of the recurrence in Eq. (9) for all query variables $q_i$ still takes $O(b^2 \cdot M \log_b M)$ time. However, we note that the calculation of $\nabla_{q_i}\phi(R_x)$ and $\nabla_{q_j}\phi(R_x)$ for any two leaf nodes $i, j \in \ell(R_x)$ shares some common intermediate calculations corresponding to the shared segment of the two paths that connect the root node to $i$ and $j$. This indeed gives rise to the notion of *common optimal substructure* which is the hallmark of problems that can be efficiently solved by dynamic programming. To this end, we propose a dynamic programming

algorithm that computes $\nabla\Phi$ in Eq. (11) in $O(M \cdot b^2)$ time by traversing the signal hierarchy tree in two passes: a bottom-up pass followed by a top-down pass. Essentially, the former computes the energy function $\phi(\cdot)$ while the latter calculates the attention vectors $\nabla_{q_i}\phi(\cdot)$ for all $i \in \ell(R_x)$. Algorithms 1–3 illustrate these steps. For the formal correctness results as well as further practical details for our proposed algorithm, see Appendix E.

## 5 Experimental Results

In this section, we present an empirical study aiming at two main goals: (1) showing the capability of the HSA mechanism in incorporating useful domain hierarchy knowledge into training better transformer models from scratch, and (2) demonstrating the unique capacity of HSA as post-training approximation of the Softmax attention in pre-trained transformer models in order to reduce the self-attention computation FLOPS in a zero-shot manner.

### 5.1 Hierarchical Language

Despite its unimodality, natural language data often comes in a semantically meaningful hierarchy (*e.g.* sections, paragraphs, sentences, etc.) which can be seen as granular abstraction of the underlying semantics in the data. Nonetheless, most transformer-based frameworks ignore this hierarchical structure which not only discards valuable prior knowledge about the semantics of the text, but in the long context scenario, it can also result in loss of information due to truncation (which is a common practice for long sequences in order to manage the computational complexity of the Softmax attention). HSA avoids truncation for long sequences by effectively reducing the computational complexity via incorporating the hierarchical abstraction.

For our empirical assessment, we have chosen the text classification problem for the sentiment analysis task on two datasets: *IMDB* [57, 1], and *Elec* [59, 2]—for sentiment classification in movie reviews and Amazon electronics product reviews, respectively [3]. The rationale for choosing these datasets lies in their inclusion of lengthy texts, which means they can benefit from hierarchical representation. For details, see Appendix I.1.

**Signal Hierarchy**: We represent each text datapoint in our datasets as a 3-level signal hierarchy: paragraphs, sentences and tokens. The position embedding at each level is the 1D grid embedding materialized by random Fourier features [48]. The tokens form the leaves of each signal hierarchy and are represented via vector embeddings. We have experimented with two token-embeddings in our experiments: the simple *Word2Vec* [60, 4], and the richer, transformer-based *T5* [70, 5].

**Experimental Settings**: We have used similar architectures for both the baseline and the HSA, each amounting to 1.2M trainable parameters. For a fair comparison, we have used the same training hyper-parameters for both models. See Appendix I for the details of experimental settings.

| Dataset | Model | Word2Vec embedding | | T5-small embedding | |
| --- | --- | --- | --- | --- | --- |
| | | Acc | F1 Score | Acc | F1 Score |
| IMDB | FSA | $0.6739 \pm 0.0004$ | $0.6739 \pm 0.0004$ | $0.7577 \pm 0.0024$ | $0.7577 \pm 0.0024$ |
| | HSA | **$0.7469 \pm 0.0029$** | **$0.7468 \pm 0.0027$** | **$0.8129 \pm 0.0010$** | **$0.8129 \pm 0.0010$** |
| Elec | FSA | $0.7182 \pm 0.0001$ | $0.7182 \pm 0.0001$ | $0.8212 \pm 0.0014$ | $0.8212 \pm 0.0014$ |
| | HSA | **$0.7549 \pm 0.0005$** | **$0.7549 \pm 0.0005$** | **$0.8521 \pm 0.0022$** | **$0.8521 \pm 0.0022$** |

Table 1: The sentiment classification Accuracy/F1 score comparison for the Flat Self-Attention (**FSA**), *i.e.* the Softmax attention, and the Hierarchical Self-Attention (**HSA**).

**HSA vs. Flat Self-Attention**: Table 1 depicts the test Accuracy and F1 Score of sentiment classification for the two models on the IMDB and Elec datasets. As these results show, HSA consistently and significantly outperforms the standard Softmax self-attention across the datasets as well as the token-embeddings. The superiority of HSA over the standard self-attention can be attributed to two main factors: (1) by incorporating the semantic hierarchical knowledge of the problem within the attention computation process, HSA effectively employs a form of regularization based on the scale separation prior that protects it against potential overfitting, and (2) for long input sequences, unlike the standard self-attention mechanism, HSA can evade truncation of the input sequence by effectively reducing the memory and the compute footprints of the attention mechanism.

**Word2Vec vs. T5 embedding**: From Table 1, we also observe that the classification results significantly improve for both models by replacing the basic Word2Vec token embedding with the richer T5

embedding. This is not surprising, but it also shows that our proposed HSA framework can be incorporated as a (shallow) adaptor on the top of pre-trained foundational models and adapt them for a new domain. Furthermore, we can see the gap between the HSA and the standard self-attention intensifies for simpler token embeddings. In other words, where we do not have access to pre-trained embedding models, the superiority of HSA and its hierarchical inductive bias is even more significant. This points to the potential significant boost we can gain by training HSA-based, multi-modal foundational models instead of the classical transformers. Due to its demanding computational requirements, we leave this empirical investigation for future work. Nonetheless, in Appendix J, we have experimented with training the HSA-based transformer from scratch (as opposed to on the top of a pre-trained embedding) and showed superior generalization capability compared to the classical transformer.

## 5.2 Multi-modal News Classification

In order to showcase the capabilities of our proposed framework in multi-modal settings, we have performed experiments for the news classification task on N24News dataset [93], where for each news article not only we have language and image modalities present, but the text itself consists of multiple sub-modalities, *i.e.* headline, abstract, image caption and body.

| Model | Acc | F1 Score |
|---|---|---|
| FSA | $0.7921 \pm 0.0036$ | $0.7902 \pm 0.0003$ |
| DeepSet | $0.7578 \pm 0.0096$ | $0.7590 \pm 0.0065$ |
| HSA | **0.7952±0.0155** | **0.8091±0.0102** |

Table 2: Accuracy/F1-score comparison for the Flat Self-Attention (**FSA**), *i.e.,* the Softmax attention, DeepSet[102], and the Hierarchical Self-Attention (**HSA**) on N24News dataset.

| Dataset | Original RoBERTa | | | | HSA-RoBERTa | | | |
|---|---|---|---|---|---|---|---|---|
| | Acc↑ | Pre↑ | Rec↑ | FL(M)↓ | Acc↑ | Pre↑ | Rec↑ | FL(M)↓ |
| IMDB(264) | 0.9558 | 0.9558 | 0.9558 | 214.94 | 0.9494 | 0.9501 | 0.9494 | **4.32** |
| AGNEWS(54) | 0.9469 | 0.9469 | 0.9469 | 8.99 | 0.9422 | 0.9423 | 0.9422 | **0.8357** |
| CoLA(12) | 0.8150 | 0.8348 | 0.8017 | 0.4441 | 0.7687 | 0.7608 | 0.7821 | **0.1912** |
| SST-2(26) | 0.9403 | 0.9404 | 0.9402 | 2.08 | 0.9025 | 0.9083 | 0.9014 | **0.4132** |
| MRPC(55) | 0.9117 | 0.9006 | 0.8938 | 9.33 | 0.8553 | 0.8613 | 0.7963 | **0.8481** |
| RTE(70) | 0.7833 | 0.7870 | 0.7796 | 15.11 | 0.7400 | 0.7400 | 0.7377 | **1.29** |
| QNLI(38) | 0.9267 | 0.9267 | 0.9268 | 4.45 | 0.5072 | 0.3398 | 0.7531 | **0.5643** |

Table 3: The FLOPs comparisons for zero-shot HSA approximation of RoBERTa-base layers 7,9,11 and RoBERTa-large layers 16,18,20,22,24 (for IMDB). We have reported MFLOPs per impacted layers as well as Accuracy (Acc), Precision (Pre) and Recall (Rec). The FLOPs are computed based on the average seq. length (shown in parentheses) for each dataset.

**Baselines**: For N24News dataset, most approaches in the literature concatenate a subset of the text sub-modalities and use that as the representation of the whole article. There are also a few multi-modal methods that incorporate the image modality as well, the best of which achieves 91% Accuracy and 90% F1 Score using 211M trainable parameters [93], not to mention incorporating other tricks such as using multiple loss functions to achieve the SOTA performance. For our experimental evaluation of HSA, however, we would need to keep these other contributing factors out, and instead compare moderate size models within our computational budget that are only different in their attention mechanisms. To this end, for our baseline method, we concatenate headline, abstract and body into one text sequence and use that to train a classical transformer (realized via one-level signal hierarchy). As the second baseline, we incorporate a multi-modal model based on the DeepSet architecture [102] to incorporate the image modality as well as the text; see Appendix I.2 for details. For all baselines as well as our HSA-based model, we ensure the number of trainable parameters is around 12M.

**Signal Hierarchy**: For the HSA-based model, each news article is represented as a signal hierarchy where at the top level the image modality as well as the text sub-modalities are represented by the *key-value* signal type (see Appendix C). The headline, abstract and caption sub-trees are further divided into tokens in the next level using the 1D Grid signal type; whereas, the body is divided into paragraphs (again using 1D Grid signal) where each paragraph is treated as a leaf by pooling the text embedding of the whole paragraph. To embed the text components at the leaves, we use *e5-base* [93, 6]; whereas, for image leaves, we use *VIT* [26, 7]. Both of these models have shown superior performance in various benchmarks [62, 77].

**Results**: Table 2 shows the test accuracy and F1 Score for the three competing methods for the N24News multi-class classification problem. From these results, we can see that our HSA methodology outperforms the baselines and the difference is significant. Interestingly, despite incorporating the additional modality of image, the performance of DeepSet significantly declines compared to the

vanilla uni-modal, flat attention. This signifies the fact that it is not enough to only incorporate other information modalities within the model, but also *how* they are incorporated is equally important to boost the model's generalization. In that sense, our proposed nested signal formalism along with its hierarchical attention mechanism provide a principled methodology to incorporate different information modalities within a transformer model.

### 5.3  Zero-Shot Hierarchical Approximation

An important feature of our proposed framework is that Theorem 3.2 gives us the theoretical basis for approximating Softmax attention via HSA given an appropriate hierarchical structure. This means that HSA can seamlessly replace regular Softmax attention *after* training, and depending on the task and the original model, the accuracy may not experience significant drop. The main objective for such replacement post-training is to reduce the number of FLOPs needed for the self-attention operation. To further examine this idea, we have adopted the classical pre-trained RoBERTa model [52] and have replaced the Softmax self-attention operation in it with HSA, and then run it against some benchmark classification datasets. During this experimentation, we made a few insightful observations. First, in general, the performance drops significantly if we replace Softmax attention with HSA for *all* hidden layers of RoBERTa, and some amount of fine-tuning is needed to regain the original performance. However, zero-shot replacement is still feasible if only a subset of layers go through HSA replacement. In particular, earlier layers seem to be more sensitive to HSA approximation while the final layers are more amenable to it. Furthermore, we observed that by interleaving HSA layers and regular Softmax layers, we can significantly reduce the accuracy gap.

Based on these observations, we applied HSA approximation to layers 7, 9 and 11 in RoBERTa-base and 16, 18, 20, 22 and 24 in RoBERTa-large. As for the hierarchy, instead of using the sentence/paragraph/etc. structures in text, we opted to fixed hierarchies generated by non-overlapping hopping windows on the input text. In particular, we used a four level hierarchy where the layers' branching factors from top to bottom are 16, 8, 4 and 2. For more experimental results on different hierarchy structures and different HSA layer combinations, see Appendix L. Table 3 compares HSA-equipped RoBERTa (henceforth HSA-RoBERTa) and the original RoBERTa in terms of FLOPs as well as Accuracy on 5 GLUE benchmarks [92], IMDB benchmark [58] and AGNEWS benchmark [104]. As these results show HSA layers significantly reduce the number of FLOPs for attention computation, and depending on the task the accuracy drop can be minimal. Keep in mind these results are obtained *completely zero-shot without any fine-tuning*. Indeed fine-tuning can further close the accuracy gap while maintaining the performance gain by HSA. This points to another HSA's strong potential: to be used as a self-attention approximation technique for long-context problems. We leave the further exploration of this direction to future work.

## 6  Conclusions

In this paper, we propose HSA, a novel mathematical framework for generalizing classical Softmax self-attention mechanism to hierarchical problems that not only occupy multiple scales but may be also defined on multiple geometries. Unlike many existing work that approach these problems via heuristic neural architectures, we mathematically derive our formulation from the principle of entropy minimization given the (nested) data signal is seen as a statistical mechanical system. Given its strong theoretical and algorithmic properties, we empirically showed that HSA can be used to inject hierarchical domain knowledge into training of transformer models and hence produce models with better generalization. We further showed that HSA can be used as a self-attention approximation technique for pre-trained models to significantly reduce the FLOPs needed for self-attention at the test time. This opens the door for HSA to be used in long context scenarios, even after training.

One high-impact future application of HSA is training large-scale foundational models that can naturally handle multi-modal and hierarchical inputs using the HSA formalism. On the theoretical side, HSA can be also extended to include non-Softmax attention mechanisms (See Appendix K). The other important future direction is application of HSA to transformer decoder for hierarchical auto-regressive generation. This is important specially because it has the potential to boost LLMs in terms of both generalization (by incorporating hierarchical, multi-modal domain knowledge) and speed (due to the low-rank nature of HSA computation). Due to its significance, we have laid the foundations of hierarchical decoding via HSA in Appendix G.

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

# Contents

## A    Notations

Table 4 summarizes our notations in the main paper. Moreover, Fig 3 visually demonstrates some of our tree-related notations.

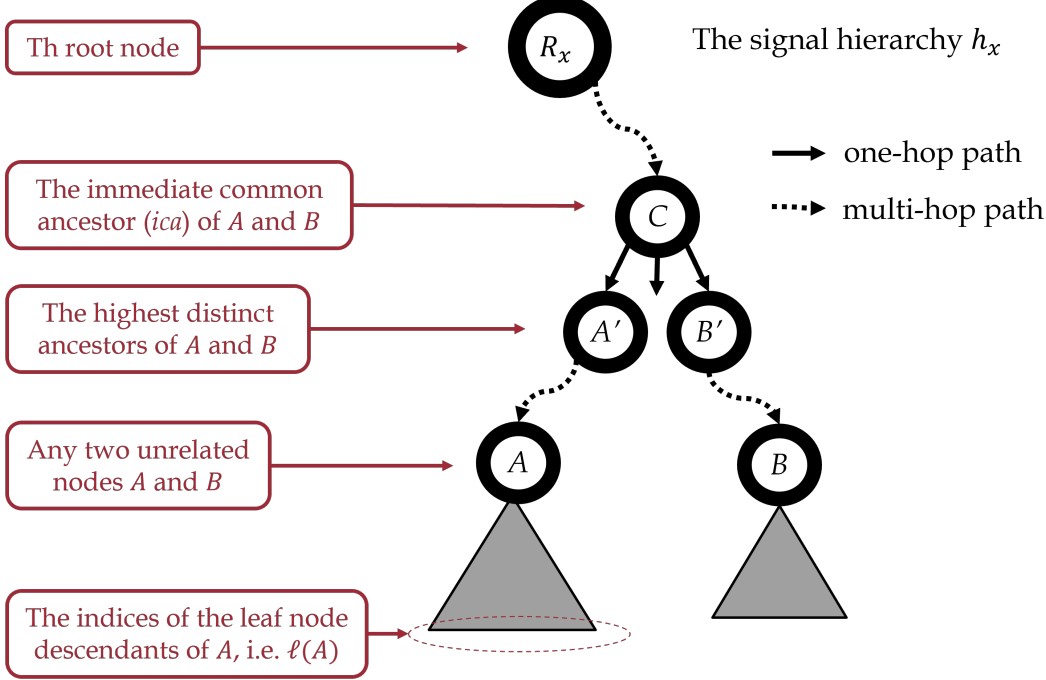

Figure 3: The visual demonstration of some of our tree-related notations in the paper.

## B    Related Work

**Hierarchical models**: The notion of hierarchy has played a key role in data representation and clustering in Machine Learning [63, 79]. In the context of transformers, the idea of multi-scale attention has been mainly used to combat the long-context challenge in language [39, 67, 98, 64, 97, 51], but it has also made its way into vision [54, 105, 53], audio [99] and graphs [86]. Nevertheless,

| Notation | Description |
|---|---|
| **Bold face** | A vector or a matrix when it is not obvious from the context |
| $\mathbf{0}$ | A column vector of zeros |
| $\mathbf{1}$ | A column vector of ones |
| $x$ | A simple or nested signal |
| $x(u)$ | The value of signal $x$ at position $u$ |
| $\Omega$ | The signal domain |
| $\mathcal{C}$ | The vector space containing the signal range |
| $d$ | The dimensionality of $\mathcal{C}$, *i.e.* the channel dimension |
| $\mathcal{D}$ | The set of all signal domains present in the problem |
| $\varepsilon_\Omega$ | The position embedding function for domain $\Omega$ |
| $c$ | The dimensionality of each position embedding |
| $\varepsilon$ | The set of all position embedding functions for all domains in $\mathcal{D}$ |
| $\mathcal{S}$ | The set of all possible simple signals in the problem |
| $\mathcal{N}_\ell$ | The set of all possible nested signals up to depth $\ell$ |
| $\mathcal{N}$ | The set of all possible nested signals in the problem |
| $\mathcal{N}_0$ | An equivalent notation for $\mathcal{C}$ |
| $h_x$ | The signal hierarchy representing the finite nested signal $x$ |
| $A, B, C, ...$ | The nodes in the signal hierarchy $h_x$ |
| $L_i$ | The leaf node in the signal hierarchy $h_x$ corresponding to the query variable $q_i$ |
| $R_x$ | The root node of the signal hierarchy $h_x$ |
| $\ell(A)$ | The set of indices of the leaf node descendants of node $A$ |
| $chd(A)$ | The children of node $A$ |
| $pa(A)$ | The parent of node $A$ |
| $sib(A)$ | The siblings of node $A$ |
| $ica(A, B)$ | The immediate common ancestors of the unrelated nodes $A$ and $B$ |
| $A', B'$ | The highest distinct ancestors of the unrelated nodes $A$ and $B$ |
| $M$ | The number of non-leaf nodes of the signal hierarchy $h_x$ |
| $b$ | The maximum branching factor of the signal hierarchy $h_x$ |
| $\lvert \cdot \rvert$ | Set cardinality |
| $\mathrm{H}(Q \mid K)$ | The conditional entropy of the query variable $Q$ given the key variable $K$ |
| $\psi_{A \to B}$ | The (directional) interaction energy between the unrelated nodes $A$ and $B$ |
| $\phi(A)$ | The energy of node $A$ |
| $\nabla_{q_i}\phi(A)$ | The gradient of the energy of node $A$ wrt the query vector $q_i$ |
| $\theta_{i,j}$ | The (directional) attention weight between query $q_i$ and key $k_j$ |
| $\Theta$ | The attention matrix |
| $\mathcal{B}$ | The set of (hierarchical) stochastic matrices respecting the block constraint wrt $h_x$ |

Table 4: The notations used in the main paper.

most of these frameworks deal with a single modality that occupies the same geometry, just at different scales. Our proposed framework, in contrast, can incorporate an arbitrary number of mutually-inconsistent geometries within its representation of the multi-scale data. Another related line of work is based on *hierarchical matrices* [35, 36] that have been used traditionally for clustering [84] as well as transition matrix approximation [8], but more recently for attention matrix [110].

**Multi-modal models**: Multi-modality has been vastly explored in Machine Learning [11] and more recently within various neural architectures, using various *fusion* techniques [30, 12, 34, 82]. As for multi-modal transformers [96], most frameworks are tailored toward a fixed set of modalities, *e.g.* vision-language [44, 40, 55, 109], audio-visual [87], audio-language [23], graph-language [103], vision-pose-audio [71], audio-vision-language [88], etc. The fusion of different modalities in these frameworks typically takes place via a heuristic operation at the embedding or the attention stages resulting in distinct architectural variants, which are typically categorized as (1) single-stream (*e.g.* [47]), (2) multi-stream (*e.g.* [55]), and (3) hybrid-stream (*e.g.* [50]). However, most of these frameworks either ignore the geometrical (positional) information for some of the input modalities, or impose artificial restrictions on input geometries such as alignment.

**Geometric Deep Learning**: Geometric Deep Learning [17] studies the invariance and equivariance properties of deep learning models by introducing the notion of *signal* and its geometry which is

explicitly modeled via the signal's domain. We build our framework also based on the same notion of signal and generalize it further to *nested signals* which can represent hierarchical, multi-modal data which potentially encompass *multiple domains*. Also, most frameworks within Geometric Deep Learning achieve the desired equivariance properties through the model's architecture (*e.g.* CNNs [49], GNNs [95], and Group-equivaraint CNNs [29]). A prominent exception is the LieTransformer [42] where the desired group-equivariance is achieved by explicit modeling of the position information and its separate similarity computation (as opposed to adding it to the feature vectors). The formulation of the position information in our framework is in part inspired by the LieTransfomer.

**The theoretical foundations of self-attention**: Despite its revolutionary success in Deep Learning, there has been quite little effort to understand the theoretical foundations of self-attention. These efforts provide various interpretations of self-attention, including the probabilistic view [80, 28], the causal view [75], the structural inference view [81], the dynamical system view [41, 56, 27], the statistical mechanical view [74], the variational denoising view [66], the clustering view [31], and the Hopfield network view [73]. In this paper, we provide a statistical mechanical perspective to derive self-attention from the first principle of entropy minimization; in that sense, our interpretation is closely related to the statistical mechanical, denoising and Hopfiled network views. More importantly, our interpretation lends itself to straightforward generalization to the hierarchical self-attention mechanism which, as we show, is both theoretically optimal and efficiently computable.

## C   Generalizing The Notion of Signal

In standard Geometric Deep Learning, signals typically represent data structures in Computer Vision, Audio Processing, Natural Language Processing and Graph and Manifold Processing. But the notion off signal is quite versatile and can be generalized to include feature representations in classical Machine Learning. In particular, we note the special case where the signal domain $\Omega$ is a countable, discrete set with no additional structure. In this case, if the elements of $\Omega$ are conceptually indistinguishable, then any signal $x$ on $\Omega$ is said to be defined on an *unordered set* and subsequently, the position embedding $\varepsilon_\Omega$ maps all the elements of $\Omega$ to the constant vector $\mathbf{0}$. The latter conveys that there is no positional information associated with the signal. As an example, a vector set can be seen as a signal defined on an unordered set.

On the other hand, if the elements of $\Omega$ are distinguishable, we can define a bijective position embedding $\varepsilon_\Omega$ to carry that information into the position vector space. We refer to signals defined on such $\Omega$ domains as *key-value* signals. For instance, a tabular feature vector in classical Machine Learning can be seen as a set of key-value pairs where the keys are the feature names and the values are the feature values, and hence modeled as a key-value signal. In this case, a text embedding model can be used to map the feature names into a vector space and regard the results as the position embeddings of those features. In other words, the notion of signal in our work is quite generic and encompasses not only the signal types in Geometric Deep Learning but also the classical tabular feature vectors.

## D   The Emergence of The Value Projection Matrix

The derived formulation for Softmax attention in (6) deviates from the classical Softmax attention in that it lacks separate *value* projections, which can be quite restrictive as it significantly reduces the model's degrees of freedom. Nevertheless, the value projections can be theoretically injected into our derived formulation by considering learnable step-size for the gradient update in (6). In particular, instead of setting step size to $\lambda = 1$, we can let $\lambda = W_v$ where $W_v \in \mathbb{R}^{d \times d}$ is a trainable parameter. By doing so, (6) changes to:

$$q_i \leftarrow q_i + \sum_{j=1, j \neq i}^{N} \frac{\exp(q_i^T k_j / \sqrt{d} + e_i^T e_j)}{\sum_{t=1, t \neq i}^{N} \exp(q_i^T k_t / \sqrt{d} + e_i^T e_j)} \cdot W_v k_j, \, 1 \leq i \leq N \qquad (14)$$

By defining $v_i = W_v k_j = W_v W_k x_j$, we effectively arrive at separate value projections, where $W_v W_k$ can be seen as the value projection matrix used in the standard Softmax attention formulation.

Note that by introducing learning step-size in the form of projection matrix, we effectively *project* the direction of the gradient vector into a new direction. So in that sense, (14) is no longer a strict

gradient ascent update. In other words, depending on the learned projection matrix $W_v$ and the value of gradient vector for point $q_i$, we may decrease or even increase the upper-bound on the conditional entropy. This extra degree of flexibility indeed enables the transformer model to best adapt to the end task. And therefore, we have adopted separate value projections in our code as well as all of our reported experiments, similar to the standard transformer architecture.

# E    Algorithmic Details

In Section 4, we proposed an efficient algorithm based on dynamic programming for the calculation of HSA. In this section, we provide formal correctness results and complexity analysis for this algorithm. We, furthermore, introduce an extension of the proposed algorithm to a more generic algorithmic template with a replaceable *black-box* base attention computation module. Finally, we discuss some practical details on how to implement our proposed algorithm on modern GPUs.

## E.1    Correctness and Complexity

First off, it is not hard to show that for the case of flat hierarchy, Algorithms 1–3 reduce to the standard Softmax attention calculations. In other words, the standard Softmax attention calculation is a special case of our proposed algorithm here. However, showing the correctness and the complexity of Algorithms 1–3 for the general case, is more involved, which we achieve through the following theorem.

**Theorem E.1.** *For a given signal hierarchy $h_x$, if both query and key variables are normalized via the LayerNorm function, then Algorithms 1–2 compute $\{\nabla_{q_i}\phi(R_x) \mid i \in \ell(R_x)\}$ in Eq. (9) in $O(M \cdot b^2)$ based on the interaction energy function defined in Eq. (7), where $b$ and $M$ are the branching factor and the number of families in $h_x$, respectively.*

*Proof.*    See Appendix H.3.    □

Once the attention values are computed using Algorithm 1, the query vector representations at the leaf nodes can be updated via the residual connection:

$$q_i \leftarrow q_i - |\ell(R_x)|\sqrt{d} \cdot \nabla_{q_i}\phi(R_x), \forall i \in \ell(R_x) \tag{15}$$

And that would conclude the HSA operation.

## E.2    Black-box Attention Computation

It is important to note that Lines 16-18 in Algorithm 3 perform the standard Softmax attention mechanism on the members of a family that contains node $A$. In other words, our proposed HSA algorithm can be seen as a *divide-and-conquer* algorithm where the attention computation on the whole sequence (*i.e.* the hierarchy's leaves) is broken down into attention computation on the much smaller families in the hierarchy (aka the *sub-problems*) via the bottom-up part of the algorithm, and then these intermediate results (aka the *sufficient statistics*) are combined through the top-down part of the algorithm to produce the final self-attention output. From this perspective, if the average branching factor (i.e. the family size) in the hierarchy is $b$, then on average, the sub-problem attention calculation takes $O(b)$ time and memory for each node $A$, which makes the $O(b^2)$ complexity for the entire family. Then intuitively for the total of $M$ families in the hierarchy, the final computational complexity comes to $O(M \cdot b^2)$. As a special case, for flat hierarchies where there is only $M = 1$ family of size $b = N$ (*i.e.* the sequence length), the complexity becomes $O(N^2)$.

More importantly, from the practical perspective, the divide-and-conquer view of the proposed algorithm encapsulates the sub-problem self-attention computation (in Lines 16-18 in Algorithm 3) as a *black-box* module that can be easily replaced by any exact or approximate function that computes the standard Softmax attention. This has a significant practical implication, as it allows the HSA algorithm to invoke any efficient attention computation frameworks in the literature as its base attention calculation sub-module. For instance, the quadratic factor $b^2$ in $O(M \cdot b^2)$ can be further reduced to linear if one employs one of the many approximation techniques proposed for efficient computation of Softmax attention [21, 13, 68, 43] as the black-box sub-problem attention computation module in Lines 16-18 in Algorithm 3.

### E.3  GPU Implementation

The Algorithms 1–2 are technically classical tree-traversal algorithms which are typically not fit for parallel processing on GPU. Indeed, this would introduce a practical challenge for incorporation of HSA within modern Deep Learning frameworks. To address this challenge, in this section, we present two major techniques for introducing parallelization both at the node level for one signal hierarchy as well as at the batch level across multiple signal hierarchies.

First, we note that all the summations in Algorithms 3 and 2 can be done in parallel for different sets of nodes in $h_x$. In particular, if a summation statement can be parallelized for $K$ nodes of $h_x$, it can be implemented as a (sparse) matrix by dense vector multiplication $\boldsymbol{W}v$, where $\boldsymbol{W} = [w_{i,j}]_{K \times S}$ is the sparse *coefficient matrix* and $v = [v_i]_{S \times 1}$ contains the values of the input terms. In particular, $w_{i,j}$ is the weight of the $j$th term for computing the summed quantity at the $i$th node (typically 1 or 0). As for the quantities in Algorithms 3 and 2, $\mu_k(\cdot)$, $\mu_q(\cdot)$ and $\eta(\cdot)$ can be parallelized over *all* the nodes in $h_x$; that is, in order to compute each one of these quantities for all nodes of $h_x$, only *one* sparse matrix-vector multiplication is needed given the appropriate coefficient matrix. The computation of $\phi(\cdot)$ and $\vartheta(\cdot)$ is also parallelizable over the nodes belonging to the same *depth* in $h_x$; in other words, given the appropriate coefficient matrices, we would need $D$ sparse matrix-vector multiplications to calculate each one of these quantities for all nodes in $h_x$, where $D$ is the depth of $h_x$. Since the coefficient matrices in this scheme are highly sparse, we have represented the coefficient matrices using sparse tensors and used the efficient implementation of sparse matrix by dense vector multiplication in Pytorch to carry out the tree-based summations in Algorithms 3 and 2.

The other fundamental aspect of parallelization in Deep Learning is batch computation, which typically boils down to matrix operations for the standard batches of fixed-size tensors. However, in our scenario, the signal hierarchies in each batch are trees with different structures as well as potentially different signal types/modalities appearing in arbitrary arrangements for each signal hierarchy in the batch. This effectively makes the classical batch computation impossible for signal hierarchies. To address this challenge, we propose a completely different technique for batch parallelization. As explained above, we already have a method to parallelize the computations within each signal hierarchy; we can further parallelize the computations across different signal hierarchies in a batch by making them part of *one* hierarchy. In particular, we introduce a *dummy root* node and make each signal hierarchy in the batch a direct child of it. The position embedding for this dummy root is set to unordered-set embedding; that is, no position embedding. This way, we end up with only one, wide signal hierarchy in our batch that is just one level deeper than the deepest signal hierarchy in the original batch. By performing the parallel version of Algorithms 1–2 (as described above) on this one "concatenated" signal hierarchy, we effectively compute all the targeted quantities for all signal hierarchies in the batch at the same time. We refer to this batch processing technique as *breadth-wise tree concatenation*.

## F  Hierarchical Transformer Encoder

The proposed HSA mechanism does not introduce any trainable parameters on its own; it is simply an attention operation. However, similar to classical transformers, we can add trainable linear projections before performing HSA. This gives rise to the *hierarchical transformer encoder (HTE)* architecture which is capable of operating on signal hierarchies representing finite nested signals. Similar to classical transformers, we also add multiple heads as well as point-wise linear projection of the output of HSA followed by some non-linearity. The same way the classical transformer layers do not change the query sequence length or the position embeddings of its tokens, HTE layers do not alter the structure of the hierarchy tree or its nodes' positional embeddings[2]. Figure 4 depicts our proposed architecture for each HTE layer.

Aside from HSA, HTE is different from classical transformer encoder in two ways. First, the LayerNorm operation is performed after linear projection as opposed to before it. As mentioned before, by doing so, the attention operation will minimize a proper energy function which is in turn a proxy for minimizing the entropy of the representation. Second, unlike simple signals in standard transformers, signal hierarchies can contain different modalities and signal domains within their

---

[2]Even though, the same position embeddings are fed to each layer, in our implementation, we have designed a separate linear projection per position embedding type per layer to project the position embeddings before the HSA operation.

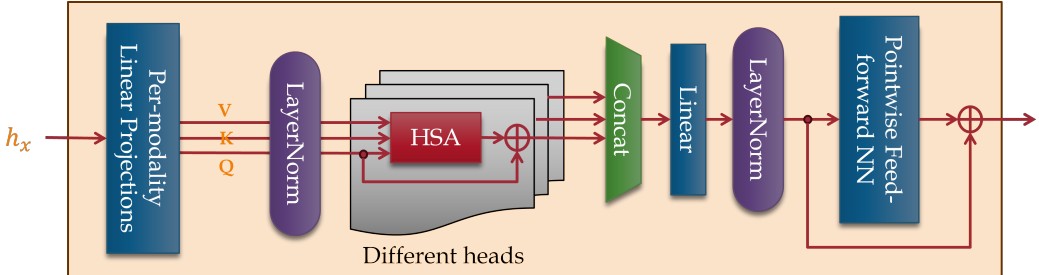

Figure 4: The proposed Hierarchical Transformer Encoder (HTE) layer architecture.

different families across the hierarchy. Therefore, using the same linear projection layer for all this various types of information may be an over-simplification. To this end, our proposed framework allows different linear projection per the type of input information. For example, the leaf vectors coming from the language and vision modalities can be transformed using their own separate linear projection layers. Note that this distinction is only allowed at the linear projection layer; the HSA operation itself is universal and does not treat different types of information differently. Also, it is assumed the different types of information in a given problem (including modalities and signal domains) are a priori known and fixed, even though each signal hierarchy in the input dataset can be an arbitrary, variable-depth composition of these known types. By making this assumption, we can know ahead of time how many linear projection layers are needed within each HTE layer.

The HTE layers can be cascaded to form a hierarchical transformer based on the HSA. Furthermore, different types of *pooling* operations can be introduced to (gradually) coarsen the hierarchical structure of the input nested signal. In particular, using a *local* pooling operation, the leaf nodes of the input signal hierarchy are either merged together or completely pooled into their parents resulting in a coarser representation of the underlying nested signal. Furthermore, since the channel dimensionality $d$ is constant across the hierarchy, *global* pooling is also well-defined which reduces the whole signal hierarchy into a single, fixed-size vector of $d$ dimensions (*e.g.* by taking the average). Depending on the application, global pooling can also be realized by taking a specific leaf node's query vector of the output signal hierarchy (*e.g.* in per-token classification tasks on uni-modal, hierarchical data).

## G  Hierarchical Auto-regressive Generation

The HSA-based, encoder-only architecture introduced in Appendix F is primarily suitable for classification and regression applications. However, for auto-regressive generation such as causal language modeling, we would need to have a decoder. One straightforward approach is to use an encoder-decoder architecture where the encoder is HSA-based while the decoder is the standard sequential decoder. In particular, in this scheme, the hierarchical self-attention is only incorporated for the initial prompt while for the generated text, we simply compute the standard flat attention. While simple, this solution does not take the full advantage HSA, especially if the generated text allows for the similar hierarchical structure as the prompt text. For instance, if the hierarchy is built upon the sentence and paragraph structure of the prompt text, then it is fairly reasonable for the generated text to have the same hierarchical construct as well. The same can be said when the hierarchy is based on fixed hopping windows over the text. In such cases, a HSA-based, *decoder-only* architecture is needed to incorporate the hierarchical structure of the generated text during auto-regressive generation.

Theoretically speaking, for a HSA-based decoder during auto-regressive generation, we would need to maintain a *dynamic* signal hierarchy where every generated token augments the signal hierarchy with at least one new leaf node and possibly multiple non-leaf nodes. Once the signal hierarchy is updated, the HSA calculations are, in principle, the same as before. Nevertheless, there are two major issues here specific to auto-regressive generation. First, unlike the HSA mechanism introduced so far, due to causal generation of tokens, leaf nodes are only allowed to attend to the other leaf nodes that have appeared *before* them; that is, we would need a *hierarchical causal masking* mechanism. Second, running the full HSA algorithm for every generated token is inefficient as it would re-compute some of the sufficient statistics in Algorithm 3, which is clearly redundant. In the following sections, we address these two problems.

## G.1 Hierarchical Causal Masking

In the standard auto-regressive generation using the self-attention mechanism, in order to prohibit tokens from attending to the future tokens, one incorporates a causal mask in calculation of the attention weights via an appropriate lower-triangular mask matrix. However, this straightforward approach will not work with hierarchical self-attention mechanism because attention weights between all tokens are *not* computed simultaneously but rather in hierarchical fashion.

Nevertheless, one can easily show that if the standard causal masking is applied at each level of the hierarchical attention calculation, at the end, no leaf token will attend to its future tokens (i.e. the tokens to its *right*) in the hierarchy. In particular, as explained in Appendix E.2, Lines 16–18 of Algorithm 3 encapsulate a black-box Softmax self-attention function that is applied for each family in the hierarchy. For applying hierarchical causal masking, we can simply apply the standard causal masking within this black-box self-attention calculation. This is equivalent to replacing the sibling function $sib(A)$ in lines 16–18 of Algorithm 3 with $sib_L(A)$ which restricts $A$'s siblings to the ones to its left (i.e. previous tokens). This simple black-box causal masking will further propagate through the hierarchy such that at the end, the leaf nodes will only attend to other leaf nodes that are located to their left. Figure 5 illustrates this process through a toy example.

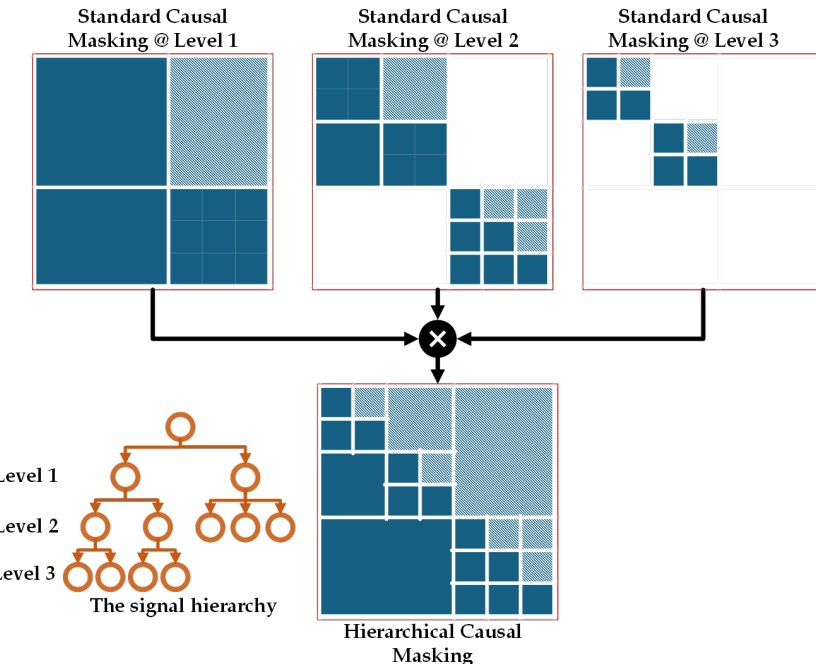

Figure 5: An illustration of the proposed Hierarchical Causal Masking scheme for hierarchical auto-regressive generation.

## G.2 Hierarchical Caching

In standard auto-regressive generation, every generated token merely attends to the tokens seen so far whose projections are cached via a *key-value cache*. This makes the attention computation for each token linear in the (generated + prompt) sequence length. This simple idea, however, is not directly applicable to the hierarchical case. One important distinction that we need to keep in mind is that in the hierarchical case, we are not only generating a sequence but also a hierarchy that comes with it; in other words, the generated sequence is the set of the leaf nodes of a hierarchy that needs to be maintained and updated as well. As such, any caching mechanism would need to maintain and update the signal hierarchy and not just its leaf nodes. Note that caching the hierarchy means maintaining its structure as well as its nodes' sufficient statistics pre-computed by Algorithm 3.

Nevertheless, during generation, we do *not* need to keep the entire signal hierarchy. In particular, in our HSA framework not every leaf node directly attend to every other leaf node; instead, leaf nodes

that are not in the same family only attend to each other at the coarse scale through their highest distinct ancestors. This means that during generation, a newly generated token (leaf node) only needs to directly attend to its previously generated leaf siblings and not other leaf nodes. Instead it will indirectly attend to other leaf nodes *en masse* by attending to their highest ancestor that is *not* an ancestor of the new token. Following this scheme, we would only need to cache a sub-tree of the original hierarchy that consists of the ancestor line of the latest generated token as well as their immediate children nodes. We refer to this sub-tree as *right-skewed* because only the right-most sibling in each family across the signal hierarchy is allowed to have children. Figure 6(A) illustrates the maximal right-skewed sub-tree for the toy hierarchy in Figure 6(B).

Once the the right-skewed sub-tree of the signal hierarchy is extracted, we can simply update as new tokens are generated. However, we have to be careful as *not all* of the newly generated tokens are added to the latest family: some new tokens may start a new family via a higher level of the hierarchy. For example, if the hierarchy for language data is built based upon the sentence and paragraph structure in the text, a new token is not always going to be part of the latest sentence or paragraph; it may start a new sentence or even a new paragraph. In such cases, more nodes need to be added to or deleted from the cache other than the new token's leaf node. These two cases are illustrated in Figure 6(C)-(D).

Finally we note that during the entire generation process the hierarchical cache remains a right-skewed tree which means that the CPU and memory complexity for calculating attention and maintaining the cache would be $O(b \log_b N)$ where $N$ is the length of the generated sequence so far and $b$ is the average branching factor of the hierarchy. This is in stark contrast to the classical key-value caching where the memory and computation are of $O(N)$ complexity, and hence shows the potential computational advantage of our hierarchical scheme.

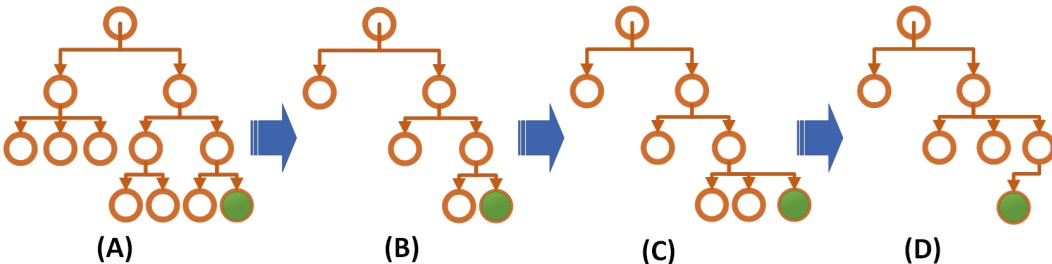

(A)  (B)  (C)  (D)

Figure 6: An illustration of the proposed hierarchical caching mechanism for hierarchical autoregressive generation: (A) The original signal hierarchy built on the prompt text. (B) The right-skewed sub-tree of the original hierarchy. (C) The updated hierarchy after generation of a new token that does not end the latest family. (D) The updated hierarchy after generation of another token that *does* end the latest family. The green leaf nodes depict the latest generated tokens in each step.

# H  Proofs

## H.1  Proposition 1: Softmax Attention

*Proof.* Since each $Q$ and $K$ variables is normalized via a LayerNorm layer, we have $\|q_i\|^2 = d$ and $\|k_i\|^2 = d$, $\forall 1 \leq i \leq N$, which would reduce the energy function in (5) to:

$$\phi(Q, K) = \sqrt{d} - \frac{1}{N} \sum_{i=1}^{N} \log \left( \frac{1}{N-1} \sum_{j=1, j \neq i}^{N} \exp \left[ q_i^T k_j / \sqrt{d} \right] \right)$$

By taking the gradient of w.r.t. $q_i$, we get:

$$\nabla_{q_i} \phi(Q, K) = \sum_{j=1, j \neq i}^{N} \frac{\exp(q_i^T k_j / \sqrt{d})}{\sum_{t=1, t \neq i}^{N} \exp(q_i^T k_t / \sqrt{d})} \cdot k_j, 1 \leq i \leq N$$

By plugging $\nabla_{q_i} \phi(Q, K)$ into (4) and setting $\tau = (N\sqrt{d})^{-1}$, $\lambda = 1$ and the sample size to 1, we will get the Softmax attention formulation in (6). $\qquad \square$

## H.2 Theorem 1: The Optimality of HSA

First off, since each $Q$ and $K$ variables is normalized via a LayerNorm layer, we have $\|q_i\|^2 = d$ and $\|k_i\|^2 = d, \forall 1 \leq i \leq N$, which would reduce the interaction energy function $\psi_{A \to B}$ in (7) to:

$$\psi_{A \to B} = -\varepsilon_\Omega(A')^T \varepsilon_\Omega(B') + \sqrt{d} - \frac{1}{\sqrt{d} \cdot |\ell(A)| \cdot |\ell(B)|} \sum_{i \in \ell(A)} \sum_{j \in \ell(B)} q_i^T k_j \qquad (16)$$

Then $\nabla_{q_i} \psi_{A \to B}$ becomes:

$$\nabla_{q_i} \psi_{A \to B} = -\frac{1}{\sqrt{d} \cdot |\ell(A)| \cdot |\ell(B)|} \sum_{j \in \ell(B)} k_j \qquad (17)$$

*Proof of stochasticity.* Next, we show that $\hat{\Theta} = -\frac{1}{\tau}\Theta$ is a stochastic matrix where $\tau = \left(|\ell(R_x)|\sqrt{d}\right)^{-1}$ and $\Theta = [\theta_{i,j}]_{|\ell(R_x)| \times |\ell(R_x)|}$ is the HSA matrix for the nested signal $x$ in (11). This is equivalent to showing that $\Theta$ is a negative matrix whose rows sum to $-\left(|\ell(R_x)|\sqrt{d}\right)^{-1}$. We prove the latter by induction on the depth of the signal hierarchy $h_x$.

**The base case**: Using the (17), for a signal hierarchy $h_x$ of depth 1 (*i.e.* a simple signal), (9) reduces to:

$$\nabla_{q_i} \phi(R_x) = -\frac{1}{\sqrt{d}|\ell(R_x)|} \left[ \sum_{L_j \in sib(L_i)} \frac{\exp\left(-\psi_{L_i \to L_j}\right)}{\sum_{L_k \in sib(L_i)} \exp\left(-\psi_{L_k \to L_j}\right)} \cdot k_j \right] = \Theta_{i.} \boldsymbol{K} \qquad (18)$$

where

$$\boldsymbol{K} = [k_1, ..., k_{|\ell(R_x)|}]^T,$$

$$\Theta_{i.} = \left[ \frac{-\exp\left(-\psi_{L_i \to L_j}\right)}{\sqrt{d}|\ell(R_x)| \sum_{L_k \in sib(L_i)} \exp\left(-\psi_{L_k \to L_j}\right)} \right]_{j=1}^{|\ell(R_x)|}$$

is the $i$th row of $\Theta$, and $L_i$, $L_j$ and $L_k$ are the leaf nodes corresponding to $q_i$, $q_j$ and $q_k$, respectively. From (18), it is clear that the elements of $\Theta$ are all negative and each row sums to $-\left(|\ell(R_x)|\sqrt{d}\right)^{-1}$.

**The induction step**: Now assume that the above statement holds for any $\Theta$ matrix derived from a signal hierarchy up to depth $T - 1$, we show that it also holds for the signal hierarchy $h_x$ of depth $T$. To this end, (9) can be written as:

$$\nabla_{q_i} \phi(R_x) = \frac{|\ell(B^i)|}{|\ell(R_x)|} \left[ \frac{\alpha(B^i) \cdot \nabla_{q_i} \phi(B^i) + \sum_{C \in sib(B^i)} |\ell(C)|\beta(B^i, C) \cdot \nabla_{q_i} \psi_{B^i \to C}}{\alpha(B^i) + \sum_{C \in sib(B^i)} |\ell(C)|\beta(B^i, C)} \right]$$

$$= \frac{|\ell(B^i)|}{|\ell(R_x)|} \left[ \mu(B^i) \nabla_{q_i} \phi(B^i) + \sum_{C \in sib(B^i)} |\ell(C)|\delta(B^i, C) \nabla_{q_i} \psi_{B^i \to C} \right]$$

where,

$$\mu(B^i) = \frac{\alpha(B^i)}{\alpha(B^i) + \sum_{D \in sib(B^i)} |\ell(D)|\beta(B^i, D)} \qquad (19)$$

$$\delta(B^i, C) = \frac{\beta(B^i, C)}{\alpha(B^i) + \sum_{D \in sib(B^i)} |\ell(D)|\beta(B^i, D)} \qquad (20)$$

and we have $\mu(B^i) + \sum_{C \in sib(B^i)} |\ell(C)|\delta(B^i, C) = 1$, for all $i \in \ell(R_x)$. On the other hand, since $B^i$ is a child of the root node $R_x$, the depth of its corresponding sub-signal hierarchy is inevitably less than $T$, and therefore its corresponding energy gradient $\nabla\phi(B^i)$ induces an attention matrix $\Theta^{B^i}$ that is negative with rows that sum to $-\left(|\ell(B^i)|\sqrt{d}\right)^{-1}$ according to the induction hypothesis. With that in mind, we can write

$$\nabla_{q_i} \phi(R_x) = \frac{|\ell(B^i)|}{|\ell(R_x)|} \left[ \mu(B^i) \Theta_{i.}^{B^i} \boldsymbol{K}^{B^i} - \sum_{C \in sib(B^i)} \frac{\delta(B^i, C)}{\sqrt{d} \cdot |\ell(B^i)|} \sum_{j \in \ell(C)} k_j \right] = \Theta_{i.} \boldsymbol{K}$$

where $\boldsymbol{\Theta}_{i.}^{B^i}$ is the $i$th row of $\boldsymbol{\Theta}^{B^i}$, $\boldsymbol{K}^{B^i} = [k_j]_{j \in \ell(B^i)}$, and we have:

$$\boldsymbol{\Theta}_{i.} = \text{concat}\left[\frac{|\ell(B^i)|}{|\ell(R_x)|}\mu(B^i)\boldsymbol{\Theta}_{i.}^{B^i}, \text{concat}\left[-\frac{\delta(B^i, C)}{\sqrt{d} \cdot |\ell(R_x)|}\mathbf{1}_{|\ell(C)|}\right]_{C \in sib(B^i)}\right]$$

Then the sum of the elements of the row vector $\boldsymbol{\Theta}_{i.}$ is given by:

$$\sum_{j \in \ell(R_x)} \theta_{i,j} = \frac{1}{|\ell(R_x)|}\left[|\ell(B^i)|\mu(B^i)\sum_{j \in \ell(B^i)} \theta_{i,j}^{B^i} - \sum_{C \in sib(B^i)} \frac{|\ell(C)|}{\sqrt{d}}\delta(B^i, C)\right]$$

$$= \frac{1}{|\ell(R_x)|}\left[-\frac{1}{\sqrt{d}}\mu(B^i) - \sum_{C \in sib(B^i)} \frac{|\ell(C)|}{\sqrt{d}}\delta(B^i, C)\right]$$

$$= -\frac{1}{\sqrt{d}|\ell(R_x)|}\left[\mu(B^i) + \sum_{C \in sib(B^i)} |\ell(C)|\delta(B^i, C)\right] = -\frac{1}{\sqrt{d}|\ell(R_x)|}$$

where the second equality comes from the induction hypothesis that $\sum_{j \in \ell(B^i)} \theta_{i,j} = -\left(|\ell(B^i)|\sqrt{d}\right)^{-1}$. In other words, $\boldsymbol{\Theta}^{R_x}$ is negative with rows that sum to $-\left(|\ell(R_x)|\sqrt{d}\right)^{-1}$, which in turn, implies that $\hat{\boldsymbol{\Theta}} = -\left(|\ell(R_x)|\sqrt{d}\right)\boldsymbol{\Theta}$ is a stochastic matrix. $\qquad\square$

Before proving the optimality of HSA, we need to show that the KL-divergence admits *optimal sub-structure* in our setting. To this end, let $\wp = [p_i]_{i=1}^N$ be a categorical distribution over $N$ items such that $\sum_{i=1}^N p_i = 1$. Furthermore, let $\mathcal{R} = \{R_1, ..., R_K\}$ be a $K$-partition on the index set $\mathcal{I} = \{1, ..., N\}$ such that $\bigcup_{j=1}^K R_j = \mathcal{I}$ and $R_i \cap R_j = \emptyset, \forall i, j \in 1..K, i \neq j$. We say a categorical distribution $\omega = [w_i]_{i=1}^N$ admits the *tie constraint* w.r.t. $\mathcal{R}$ iff we have $w_i = w_j$ if $\exists R_k \in \mathcal{R}$ s.t. $i, j \in R_k$. We refer to set of all such distributions as $W_\mathcal{R}$.

Given a distribution $\omega \in W_\mathcal{R}$ and the *sub-partition* $\mathcal{R}' \subset \mathcal{R}$, the projection of $\omega$ on $\mathcal{R}'$ is defined as $\omega_{\perp \mathcal{R}'} = [w_i/h]_{i \in \mathcal{I}(\mathcal{R}')}$ where $\mathcal{I}(\mathcal{R}') = \bigcup_{R \in \mathcal{R}'} R$, and $h = \sum_{i \in \mathcal{I}(\mathcal{R}')} w_i$ is the re-normalization constant. From this definition, it is clear $\omega_{\perp \mathcal{R}'}$ is a categorical distribution restricted to the items in the partition $\mathcal{R}'$.

**Lemma H.1** (The optimal sub-structure of the KL-divergence). *Let $\wp$, $\mathcal{R}$ and $W_\mathcal{R}$ be defined as above; furthermore, let $\omega^* \in W_\mathcal{R}$ be the closest categorical distribution in $W_\mathcal{R}$ to $\wp$ in terms of the KL-divergence; that is,*

$$\omega^* = \arg\min_{\omega \in W_\mathcal{R}} D_{KL}(\omega \| \wp)$$

*Then, for any $\mathcal{R}' \subset \mathcal{R}$, we have:*

$$\omega^*_{\perp \mathcal{R}'} = \arg\min_{\omega \in W_{\mathcal{R}'}} D_{KL}(\omega \| \wp_{\perp \mathcal{R}'})$$

*Proof.* Let us assume the closest distribution in $W_{\mathcal{R}'}$ to $\wp_{\perp \mathcal{R}'}$ is $\omega'$ that is *not* equal to $\omega^*_{\perp \mathcal{R}'}$. Then we have,

$$D_{KL}(\omega^* \| \wp) = \sum_{i \in \mathcal{I}(\mathcal{R}')} w_i^* \log(w_i^*/p_i) + \sum_{i \in \mathcal{I}(\mathcal{R} \setminus \mathcal{R}')} w_i^* \log(w_i^*/p_i)$$

$$= h_1 \log(h_1/h_2) + h_1 \sum_{i \in \mathcal{I}(\mathcal{R}')} w_{\perp \mathcal{R}' i}^* \log(w_{\perp \mathcal{R}' i}^*/p_{\perp \mathcal{R}' i}) + \sum_{i \in \mathcal{I}(\mathcal{R} \setminus \mathcal{R}')} w_i^* \log(w_i^*/p_i)$$

$$> h_1 \log(h_1/h_2) + h_1 \sum_{i \in \mathcal{I}(\mathcal{R}')} w_i' \log(w_i'/p_{\perp \mathcal{R}' i}) + \sum_{i \in \mathcal{I}(\mathcal{R} \setminus \mathcal{R}')} w_i^* \log(w_i^*/p_i)$$

$$= \sum_{i \in \mathcal{I}(\mathcal{R}')} h_1 w_i' \log(h_1 w_i'/p_i) + \sum_{i \in \mathcal{I}(\mathcal{R} \setminus \mathcal{R}')} w_i^* \log(w_i^*/p_i)$$

$$= D_{KL}(\omega'' \| \wp)$$

where

$$\omega'' = [w_i'']_{i=1}^N, \text{ such that } w_i'' = \begin{cases} h_1 w_i', & \text{for } i \in \mathcal{I}(\mathcal{R}') \\ w_i^*, & \text{for } i \in \mathcal{I}(\mathcal{R} \setminus \mathcal{R}') \end{cases}$$

and $h_1 = \sum_{i \in \mathcal{I}(\mathcal{R}')} w_i^*$ and $h_2 = \sum_{i \in \mathcal{I}(\mathcal{R}')} p_i$ are the re-normalization coefficients. The inequality in the above derivation is the direct result of the fact that $\omega'$ is the closest distribution to $\wp_{\perp \mathcal{R}'}$ in $W_{\mathcal{R}'}$. This further implies that we just found another distribution $\omega'' \in W_{\mathcal{R}}$ that is closer to $\wp$ than $\omega^*$ is. And this contradicts our assumption regarding the optimality of $\omega^*$. Therefore, $\omega_{\perp \mathcal{R}'}^*$ must be the closest distribution to $\wp_{\perp \mathcal{R}'}$ in $W_{\mathcal{R}'}$. $\qquad\square$

Intuitively speaking, Lemma H.1 states that any sub-structure of an optimal solution for the KL-divergence to a target distribution is also optimal. With that, we are now ready to show the optimality of HSA.

*Proof of optimality.* We would like to show that the HSA formulation in (9) results in a self attention matrix $\hat{\Theta}$ that minimizes the total KL-divergence in (12). In order to do so, we derive the optimal solution for the total KL-divergence and show that it obeys the recurrence in (9).

For a signal hierarchy $h_x$ rooted at $R_x$, let $\hat{\Theta}^R$ denote the closest HSA matrix in $\mathcal{B}$ (the space of all matrices that admit the block constraint according to the signal hierarchy $h_x$) to the flattened self-attention matrix $\Theta^f$ described by (13). That is,

$$\hat{\Theta}^R = \arg\min_{\Theta \in \mathcal{B}} \sum_{i \in \ell(R_x)} D_{KL}(\theta_{i,\cdot} \| \theta_{i,\cdot}^f) \equiv \arg\min_{\Theta \in \mathcal{B}} \bar{D}_{KL}(\Theta \| \Theta^f) \tag{21}$$

Since each row of $\hat{\Theta}^R$ is a categorical distribution, by applying Lemma H.1 to the rows of $\hat{\Theta}^R$, it is straightforward to see that the diagonal blocks of $\hat{\Theta}^R$ corresponding to the children of $R_x$ are also (up to a re-normalization factor) the closest HSA matrices to the restriction of the flattened self-attention matrix $\Theta^f$ to the corresponding sub-hierarchies. For the child node $A \in chd(R_x)$, the renormalized restriction of $\Theta^f$ to $A$ is denoted by $\Theta^{f,A}$. The elements of $\Theta^f$ are then can be written as:

$$\forall i, j \in \ell(R_x), \theta_{i,j}^f = \begin{cases} \frac{z_i}{z_i + \bar{z}_i} \theta_{i,j}^{f,A^i}, & \text{if } A^i = A^j \\ \frac{b_{i,j}}{z_i + \bar{z}_i}, & \text{if } A^i \neq A^j \end{cases} \tag{22}$$

where $A^i$ denotes that child of $R_x$ that contains the $i$th leaf node, $b_{i,j} = \exp(-\psi_{i \to j})$, $z_i = \sum_{j \in \ell(A^i)} b_{i,j}$, and $\bar{z}_i = \sum_{j \in \ell(R_x) \setminus \ell(A^i)} b_{i,j}$. Similarly, if we denote the renormalized restriction of $\hat{\Theta}^R$ to $A$ by $\hat{\Theta}^{R,A}$, the elements of of $\hat{\Theta}^R$ are then can be written as:

$$\forall i, j \in \ell(R_x), \hat{\theta}_{i,j}^R = \begin{cases} \mu(A^i) \hat{\theta}_{i,j}^{R,A^i}, & \text{if } A^i = A^j \\ \delta(A^i, A^j), & \text{if } A^i \neq A^j \end{cases} \tag{23}$$

where $\mu(A^i)$ and $\delta(A^i, A^j)$ are unknown coefficients. Note that unlike (22), for the case of $A^i \neq A^j$, we only have *one* number representing the attention weight between sub-trees $A^i$ and $A^j$ - *i.e.* $\delta(A^i, A^j)$. This is due to the block constraint being enforced on $\hat{\Theta}^R$. Similarly, the block constraint requires the renormalization coefficient for every child $A^i$, *i.e.* $\mu(A^i)$, to be the same for all the rows $k \in \ell(A^i)$. If we assume we already know the optimal restricted HSA matrices $\hat{\Theta}^{R,A}, \forall A \in chd(R_x)$, our goal reduces to computing the values of $\mu(A)$ and $\delta(A, B)$ for all $A, B \in chd(R_x)$ such that the

total KL-divergence in (21) is minimized. By plugging Eqs.(22) and(23) into (21), we get:

$$\bar{D}_{KL}(\hat{\Theta}^R \| \Theta^f) = \sum_{A \in chd(R_x)} \sum_{i \in \ell(A)} \left[ \sum_{j \in \ell(A)} \mu(A) \hat{\theta}_{i,j}^{R,A^i} \log \left( \frac{\mu(A)\hat{\theta}_{i,j}^{R,A^i}(z_i + \bar{z}_i)}{z_i \theta_{i,j}^{f,A^i}} \right) \right.$$

$$+ \sum_{B \in sib(A)} \sum_{j \in \ell(B)} \delta(A,B) \log \left( \frac{(z_i + \bar{z}_i)\delta(A,B)}{b_{i,j}} \right) \Big]$$

$$= \sum_{A \in chd(R_x)} \left[ \mu(A) \left( \bar{D}_{KL}(\hat{\Theta}^{R,A} \| \Theta^{f,A}) + |\ell(A)| \log \mu(A) + \sum_{i \in \ell(A)} \log \left( \frac{z_i + \bar{z}_i}{z_i} \right) \right) \right.$$

$$+ \sum_{B in sib(A)} \left( |\ell(A)||\ell(B)|\delta(A,B) \log \delta(A,B) \right.$$

$$+ \delta(A,B) \left[ |\ell(B)| \sum_{i \in \ell(A)} \log(z_i + \bar{z}_i) - \sum_{i \in \ell(A)} \sum_{j \in \ell(B)} \log b_{i,j} \right] \Big) \Big] \tag{24}$$

Where $\bar{D}_{KL}(\hat{\Theta}^{R,A} \| \Theta^{f,A})$ is the optimal value of the total KL-divergence for the sub-problem induced by the child node $A$ of $R_x$. Since $\hat{\Theta}^R$ is the minimizer of (24), the values of $\mu(A)$ and $\delta(A,B), \forall A, B \in chd(R_x)$ must be chosen such that they minimize (24). Furthermore, each row of the matrix $\hat{\Theta}^R$ must sum to 1, which results in the following set of constraints on the values of $\mu(A)$ and $\delta(A,B)$:

$$\forall i \in \ell(R_x), \sum_{j \in \ell(R_x)} \hat{\theta}_{i,j}^R = 1 \Rightarrow \sum_{j \in \ell(A^i)} \hat{\theta}_{i,j}^R + \sum_{j \in \ell(R_x) \setminus \ell(A^i)} \hat{\theta}_{i,j}^R = 1$$

$$\Rightarrow \mu(A^i) \sum_{j \in \ell(R_x)} \hat{\theta}_{i,j}^{R,A^i} + \sum_{B \in sib(A^i)} |\ell(B)|\delta(A^i, B) = 1$$

$$\Rightarrow \mu(A) + \sum_{B \in sib(A)} |\ell(B)|\delta(A,B) = 1, \forall A \in chd(R_x) \tag{25}$$

where the second line is obtained by incorporating (23) and the last line uses the fact that the rows of the restricted matrix $\hat{\Theta}^{R,A^i}$ are already normalized. To optimize (24) w.r.t. $\mu(A)$ and $\delta(A,B)$, $\forall A, B \in chd(R_x)$ while enforcing the constraints in (25), we form the Lagrangian as follows:

$$\mathfrak{L}\big(\mu(A), \delta(A,B), \lambda_A; \forall A, B \in chd(R_x)\big)$$

$$= \bar{D}_{KL}(\hat{\Theta}^R \| \Theta^f) - \sum_{A \in chd(R_x)} \lambda_A \left[ \mu(A) + \sum_{B \in sib(A)} |\ell(B)|\delta(A,B) - 1 \right] \tag{26}$$

where $\lambda_A, \forall A \in chd(R_x)$ are the Lagrange multipliers. By taking the partial derivatives of the Lagrangian w.r.t. $\mu(A)$ and $\delta(A,B)$ and solving for them, we get:

$$\mu(A) = \exp \left[ \frac{1}{|\ell(A)|} \left( \lambda_A - \bar{D}_{KL}(\hat{\Theta}^{R,A} \| \Theta^{f,A}) + \sum_{i \in \ell(A)} \log \left( \frac{z_i}{z_i + \bar{z}_i} \right) \right) - 1 \right],$$

$$\delta(A,B) = \exp \left[ \frac{1}{|\ell(A)|} \left( \lambda_A - \frac{1}{|\ell(B)|} \sum_{i \in \ell(A)} \sum_{j \in \ell(B)} \log b_{i,j} - \sum_{i \in \ell(A)} \log(z_i + \bar{z}_i) \right) - 1 \right] \tag{27}$$

Now if we plug (27) into the constraints in (25), we can solve for $\lambda_A$'s, which can be further put back into (27) to derive the values of $\mu(A)$ and $\delta(A,B)$ as:

$$\mu(A) = \frac{\gamma(A)}{\gamma(A) + \sum_{C \in sib(A)} |\ell(B)|\zeta(A,C)}, \delta(A,B) = \frac{\zeta(A,B)}{\gamma(A) + \sum_{C \in sib(A)} |\ell(B)|\zeta(A,C)} \tag{28}$$

where

$$\gamma(A) = \exp \left[ \frac{1}{|\ell(A)|} \left( \sum_{i \in \ell(A)} \log z_i - \bar{D}_{KL}(\hat{\Theta}^{R,A} \| \Theta^{f,A}) \right) \right] \tag{29}$$

$$\zeta(A,B) = \exp \left[ \frac{1}{|\ell(A)||\ell(B)|} \sum_{i \in \ell(A)} \sum_{j \in \ell(B)} \log b_{i,j} \right] = \exp(-\psi_{A \to B}) \tag{30}$$

where the last equality directly results from the definition of $b_{i,j}$ and the definition of the interaction energy between $A$ and $B$ in (7). In case $R_x$ is of depth 1 (that is, $A$ is a leaf node), $\gamma(A)$ is simply defined to be 0. By plugging these values into (24) and doing some algebra, we derive the optimal value of the total KL-divergence as follows:

$$\bar{D}_{KL}(\hat{\mathbf{\Theta}}^R \| \mathbf{\Theta}^f) = \sum_{A \in chd(R_x)} \left[ \sum_{i \in \ell(A)} \log(z_i + \bar{z}_i) - |\ell(A)| \log \left( \gamma(A) + \sum_{B \in sib(A)} |\ell(B)| \zeta(A, B) \right) \right]$$

(31)

On the other hand, using (29), we can derive $\gamma(R_x)$ as:

$$\gamma(R_x) = \exp \left[ \frac{1}{|\ell(R_x)|} \left( \sum_{i \in \ell(R_x)} \log z_i - \bar{D}_{KL}(\hat{\mathbf{\Theta}}^R \| \mathbf{\Theta}^f) \right) \right]$$

(32)

Now by plugging (31) into (32), applying (30), and taking the logarithm of both sides, we arrive at:

$$\log \gamma(R_x) = \sum_{A \in chd(R_x)} \frac{|\ell(A)|}{|\ell(R_x)|} \log \left[ \exp \left( \log \gamma(A) \right) + \sum_{B \in sib(A)} |\ell(B)| \exp(-\psi_{A \to B}) \right]$$

(33)

By comparing (33) to the definition of the energy of the signal hierarchy in (8), it is clear that our proposed energy function $\phi(\cdot)$ and $-\log \gamma(\cdot)$ follow the exact same recurrence dynamic. Furthermore, since the initial values of these two functions at the leaf nodes are both equal to $\infty$, we can conclude that $\gamma(A) = exp(-\phi(A))$ for all nodes $A$ in the signal hierarchy $h_x$. In other words, $\gamma(\cdot)$ and $\zeta(\cdot, \cdot)$ are respectively the exact same functions as $\alpha(\cdot)$ and $\beta(\cdot, \cdot)$ in (10). This further means that the optimal coefficients $\mu(\cdot)$ and $\delta(\cdot, \cdot)$ in (28) to update the optimal self-attention matrix recurrence in (23) are the exact same coefficients in our proposed recurrence in (9) to compute hierarchical self-attention. Since both methods result in the same attention matrix for the base case of one-level hierarchy (*i.e.* the standard Softmax attention), *and* also follow the exact same recurrence dynamic, we can conclude that they are equivalent. This means that our proposed HSA formulation is also optimal in the sense of the total KL-divergence, which concludes the proof. $\square$

### H.3 Theorem 2: The Correctness and The Complexity of Algorithms 1–3

*Proof.* Before proving the correctness and the complexity of our proposed algorithm, we show the complexity of directly calculating (9). In order to compute $\nabla_{q_i} \phi(R_x)$, we would need to first calculate the node energy function $\phi(\cdot)$ at every node in the signal hierarchy using the recursive formula in (8). For a signal hierarchy with $M$ internal nodes and the maximum $b$ branching factor, we would have $O(M \cdot b)$ nodes in the hierarchy, at each one of them, we would need to compute the sum in in (8) over their $O(b)$ siblings. This would make the total complexity of calculating $\phi(\cdot)$ $O(M.b^2)$. This is essentially the complexity of the recursive function in Algorithm 3.

Next, to compute $\nabla_{q_i} \phi(\cdot)$ from (9), we need to traverse the path from the root node to the leaf node corresponding to $q_i$ which has $O(\log_b M)$ nodes. In each node, we also need to calculate a sum over the $O(b)$ siblings of that node, which makes the cost of calculating $\nabla_{q_i} \phi(\cdot)$ $O(b \log_b M)$. However, since we would need to repeat this calculation for all $O(M \cdot b)$ leaf nodes $q_i$'s, the total cost of computing HSA would become $O(b^2.M \log_b M)$.

Moving on with the proof, we note that the recurrence relation in (9) can be written as:

$$\nabla_{q_i} \phi(R_x) = \exp \left( f(B^i) \right) \nabla_{q_i} \phi(B^i) + g(B^i)$$

(34)

where

$$f(B^i) = \log \mu(B^i) , g(B^i) = \sum_{C \in sib(B^i)} |\ell(C)| \delta(B^i, C) \cdot \nabla_{q_i} \psi_{B^i \to C}$$

(35)

and $\mu(\cdot)$ and $\delta(\cdot, \cdot)$ are given in (19) and (20). Furthermore, (34) is a first-order, non-homogeneous recurrence relations with variable coefficients for which we can derive the following closed-form solution:

$$\nabla_{q_i} \phi(R_x) = \sum_{B \in R_x \rightsquigarrow L_i} \left[ g(B) \exp \left( \sum_{C \in R_x \rightsquigarrow Pa(B)} f(C) \right) \right] = \sum_{B \in R_x \rightsquigarrow L_i} \left[ g(B) \exp \left( u(Pa(B)) \right) \right]$$

(36)

where

$$u(A) = \sum_{C \in R_x \rightsquigarrow A} f(C) = f(A) + u\big(Pa(A)\big), \tag{37}$$

$R_x \rightsquigarrow A$ denotes the set of all nodes on the path from the root to node $A$ *excluding* the root itself, $L_i$ is the leaf node corresponding to $q_i$ and $Pa(B)$ denotes the parent of node $B$. Furthermore, define:

$$\vartheta(A) \equiv \sum_{B \in R_x \rightsquigarrow A} \left[ g(B) \exp\left( u\big(Pa(B)\big) \right) \right] = g(A) \exp\left( u\big(Pa(A)\big) \right) + \vartheta\big(Pa(A)\big) \tag{38}$$

Then it is straightforward to see:

$$\nabla_{q_i} \phi(R_x) = \vartheta(L_i) \, , \forall i \in \ell(R_x) \tag{39}$$

On the other hand, given that each $Q$ and $K$ variables are normalized via a LayerNorm layer, we can plug (17) into (35), to get:

$$
\begin{aligned}
g(A) &= - \sum_{C \in sib(A)} \frac{\delta(A,C)}{|\ell(A)| \sqrt{d}} \sum_{j \in \ell(C)} k_j \\
&= - \frac{1}{|\ell(A)| \sqrt{d} \big[ \alpha(A) + \sum_{D \in sib(A)} |\ell(D)| \beta(A,D) \big]} \sum_{C \in sib(A)} \left[ \beta(A,C) \sum_{j \in \ell(C)} k_j \right] \\
&= - \frac{1}{|\ell(A)| \sqrt{d} \big[ \exp\left( -\phi(A) \right) + \exp\left( -\eta(A) \right) \big]} \sum_{C \in sib(A)} \left[ |\ell(C)| \beta(A,C) \rho_k(C) \right] \\
&= - \frac{1}{|\ell(A)| \sqrt{d} \big[ \exp\left( -\phi(A) \right) + \exp\left( -\eta(A) \right) \big]} \sum_{C \in sib(A)} \left[ |\ell(C)| \exp(-\psi_{A \to C}) \rho_k(C) \right] \\
&= - \frac{\sum_{C \in sib(A)} |\ell(C)| \exp\left( \varepsilon_\Omega(A)^T \varepsilon_\Omega(C) - \sqrt{d} + \frac{1}{\sqrt{d}} \rho_q(A)^T \rho_k(C) \right) \rho_k(C)}{|\ell(A)| \sqrt{d} \big[ \exp\left( -\phi(A) \right) + \exp\left( -\eta(A) \right) \big]}
\end{aligned}
\tag{40}
$$

where the last equality is derived from (16), and we have:

$$
\begin{aligned}
\eta(A) &\equiv - \log \Big[ \sum_{D \in sib(A)} |\ell(D)| \beta(A,D) \Big] \\
&= - \log \Big( \sum_{B \in sib(A)} |\ell(B)| \exp \big[ \varepsilon(A)^T \varepsilon(B) + \frac{1}{\sqrt{d}} \rho_q(A)^T \rho_k(B) - \sqrt{d} \big] \Big)
\end{aligned}
\tag{41}
$$

and

$$\rho_q(A) \equiv \frac{1}{|\ell(A)|} \sum_{j \in \ell(A)} q_j \, , \, \rho_k(A) \equiv \frac{1}{|\ell(A)|} \sum_{j \in \ell(A)} k_j \tag{42}$$

Furthermore, we can rewrite $f(A)$ as:

$$
\begin{aligned}
f(A) = \log \mu(A) &= \log \left[ \frac{\alpha(A)}{\alpha(A) + \sum_{D \in sib(A)} |\ell(D)| \beta(A,D)} \right] \\
&= \log \left[ \frac{\exp\left( -\phi(A) \right)}{\exp\left( -\phi(A) \right) + \exp\left( -\eta(A) \right)} \right] \\
&= \log \left[ \frac{1}{1 + \exp\left( \phi(A) - \eta(A) \right)} \right] \\
&= \log \mathtt{Sigmoid}\big[ \eta(A) - \phi(A) \big]
\end{aligned}
\tag{43}
$$

Finally, by plugging (43) and (40) into the recurrence relations in (37) and (38), we arrive at Lines 3–4 of Algorithm 2. This means that after completion of Algorithm 2, we can read off $\nabla_{q_i} \phi(R_x) = \vartheta(L_i)$ at the leaf nodes of the hierarchy. This proves the correctness of our proposed algorithm.

As for the complexity, since Algorithm 3 visits each $O(M \cdot b)$ nodes of the hierarchy once and performs the summation in Lines 16–18 over the $O(b)$ siblings of each node, the complexity of Algorithm 3 is $O(M \cdot b^2)$, which means the total complexity of computing HSA using our dynamic programming approach is $O(M \cdot b^2)$. And this concludes the proof. $\qquad \square$

# I  Experimental Settings

In this appendix, we detail the experimental settings used for the reported experiments in the main paper, which are all completed on 2 Nvidia Titan V GPUs with 24GB GPU memory on a local Lambda box.

## I.1  Datasets

**Hierarchical Language**: For this experiment, we have chosen the text classification problem for the sentiment analysis task on two datasets: *IMDB* [57, 1], and *Elec* [59, 2]—for sentiment classification in movie reviews and Amazon electronics product reviews, respectively. The reason behind choosing these datasets lies in their inclusion of lengthy texts, which means they can benefit from hierarchical representation. Both datasets have 2 classes. Table 5 summarizes some basic statistics for these datasets. For the validation set, we have used $10\%$ of the training set.

|      | # Classes | Train Size | Test Size | Avg. Word/Doc. |
|------|-----------|------------|-----------|----------------|
| **IMDB** | 2 | 25K | 25k | 235 |
| **Elec** | 2 | 25K | 25k | 108 |

Table 5: The statistics for the IMDB and Elec datasets used for the sentiment classification task.

**Multi-modal News Classification**: For this task, we have performed experiments for the news classification task on N24News dataset [93], where for each news article not only we have language and image modalities present, but the text itself consists of multiple sub-modalities, *i.e.* headline, abstract, image caption and main body. N24News dataset consists of total of $61,218$ news stories and $24$ total number of classes. The source of the news articles is the New York Times from 2010 to 2020. For training/validation/testing splitting, we use random splitting of ratio 8:1:1 used by the original paper.

## I.2  Model Architectures

All the competitor models in our experiments follow the same general architectural pattern: an attention layer, followed by a global pooling layer, followed by a multi-layer Perceptron (MLP).

**The attention layer**: For our main model, the attention layer is a single-layer, HSA-based transformer as depicted in Fig. 4. For brevity, we refer to this architecture simply as **HSA**. For the flattened self-attention (**FSA**) baseline, the same attention layer as Fig. 4 is applied, except that the input signal hierarchy to the layer is flattened into a one level (simple) signal (For experiments using the standard transformer layers, see Appendix J). As shown by Proposition 1, a single level signal hierarchy is mathematically equivalent to the standard Softmax attention mechanism, which means that we can view **FSA** representing the standard Softmax attention. For the **DeepSet** baseline in the multi-modal experiment, we apply the same architecture as Fig. 4 for the attention layer, except the attention operation itself is disabled. That is, all the other neural operations in the HTE layer is applied except for the attention. This effectively means that we individually transform each token in the signal hierarchy without letting them interact with each other through the attention mechanism. This operation followed by pooling and MLP layers effectively implements a DeepSet architecture [102] for combining the token representations in the input signal into a single, fixed sized vector. Note that in all of our experiments across different models, the attention layer is simply the HTE layer in Fig. 4 or a variant of it, and as such we can specify the architectural details for each experiment/model using the same hyper-parameters, as detailed in Table 6. To ensure a fair comparison, we maintain an equal number of parameters across all models within each experiment.

**The global pooling layer**: The purpose of global pooling layer is to aggregate the leaf representation across the hierarchy into a single, fixed-size vector. We have multiple options for this layer; in our experiments, we have chosen the global mean pooling.

**The MLP**: After pooling the representation into a single vector, we apply a 1-hidden layer MLP on the resulted vector, the dimensions of which are summarized in Table 6.

| Experiment | Hierarchical Language | | Multi-modal News Classification | | |
|---|---|---|---|---|---|
| Model | FSA | HSA | Deep Set | FSA | HSA |
| # of Parameters | 1.2M | 1.2M | 13.4M | 11.8M | 11.8M |
| # of Heads | 3 | 3 | 3 | 3 | 3 |
| HTE Layer Output dim | 128 | 128 | 512 | 512 | 512 |
| Position Embedding dim | 768 | 768 | 768 | 768 | 768 |
| Attention dim | 128 | 128 | 768 | 256 | 256 |
| MLP dim | 128 | 128 | 512 | 512 | 512 |

Table 6: Configuration of model architectures employed in all experiments/models

## I.3 Training Hyper-parameters

Table 7 summarizes the training hyper-parameters used for each experiment. We use the same hyper-parameters across different baselines for each experiment.

| Experiment | Hierarchical Language | Multi-modal News Classification |
|---|---|---|
| Loss Function | Standard Cross-Entropy Loss | Standard Cross-Entropy Loss |
| Train Batch Size | 64 | 512 |
| Test Batch Size | 64 | 512 |
| Optimizer | AdamW | AdamW |
| Max Tokens for Training | 512 | 512 |
| Learning Rate | $2 \times 10^{-5}$ | $1 \times 10^{-4}$ |
| Learning Rate Scheduler | LinearLR | LinearLR |
| # Train Epochs | 30 | 5 |

Table 7: The training hyperparameters used for each experiment.

## J  Comparison to The Classical Transformer Architecture

The experimental results reported in Sections 5.1 and 5.2 aimed at comparing the performance of our HSA framework vs. that of the flat attention, where the rest of the architecture aside from the attention mechanism were the same one proposed in Appendix F. However, a more practical comparison would be the one between the performance of these two mechanisms within the classical transformer architecture proposed by [90]. To this end, we have conducted experiments where we train from scratch and compare a standard RoBERTa model and a HSA-RoBERTa model (as proposed in Section 5.3 on two GLUE benchmarks. For HSA-RoBERTa, we simply replace the standard flat self-attention operation with HSA, while the hierarchy is imposed a fixed four-level hierarchy where the branching factors from bottom to to are 16, 8, 4, and 2.

| Dataset | Model | Accuracy | Precision | Recall | F1 Score |
|---|---|---|---|---|---|
| MRPC | RoBERTa | 0.8608 | 0.8872 | 0.9058 | 0.8964 |
| | HSA-RoBERTa | **0.8846** | **0.9165** | **0.9093** | **0.9129** |
| RTE | RoBERTa | 0.8158 | 0.7985 | **0.8167** | **0.8075** |
| | HSA-RoBERTa | 0.8158 | **0.8076** | 0.8015 | 0.8045 |
| QQP (after 4 epochs) | RoBERTa | 0.3681 | 0.3681 | 0.5381 | 0.4371 |
| | HSA-RoBERTa | **0.9185** | **0.8764** | **0.9065** | **0.8911** |

Table 8: The comparison of training RoBERTa vs. HSA-RoBERTa from scratch on three GLUE datasets.

Table 8 shows the results on the evaluation set of each dataset after training. As these results show, the incorporation of HSA within a standard transformer architecture not only can improve the computational complexity of self-attention computation, but it can also improve the evaluation metrics due to the regularization effects of our hierarchical framework. This result is consistent with the ones in Sections 5.1 and 5.2. Furthermore, for the QQP dataset, we have shown the results just after 4 epochs; interestingly, these results show that HSA-RoBERTa converges much faster than the standard RoBERTa model.

## K    Going Beyond Softmax Attention

One of the primary contributions of our work is generalizing Softmax attention from flat signals to the hierarchical structure of nested signals. This generalization is further confirmed by the theoretical result of Theorem 3.2. However, there has been a significant effort in the literature to explore other forms of attention mechanisms than Softmax attention [20, 22, 108, 78, 37]. One of the main motivations of departing from the Softmax attention lies in the fact that Softmax attention induces *dense* probability distribution over all tokens. Sparse attention [20, 22], on the other hand, organically induces sparse probability distributions over tokens which can greatly improve the interpretability and computational efficiency of transformer models. A natural question is then whether our hierarchical derivation can be applied to other forms of attention, in particular the sparse attention. In other words, can our formalism also generalize sparse attention from flat signals to the hierarchical structure of nested signals?

### K.1    Sparse Attention as Energy Minimization

The first step toward generalizing Sparse attention to the hierarchical setting is to formulate the flat case as an energy minimization problem, much like what we did in Proposition 3.1 for the Softmax attention. To this end, we would need to define an appropriate energy function for the sparse attention. But before that let us define a generic form of energy function that can encompass various forms probability-based attentions.

Let $Q$ and $K$ be sets of query and key vectors with bounded norms (*e.g.* induced by LayerNorm) respectively; we define the generic energy function as:

$$\phi^g(Q, K) = -\frac{1}{N} \sum_{i=1}^{N} \phi_i^g(z_{i1}, z_{i2}, \ldots, z_{iN}), \text{ where } z_{ij} = q_i^T k_j \tag{44}$$

Then the gradient of $\phi^g(Q, K)$ w.r.t the query token $q_i$ is:

$$\nabla_{q_i} \phi^g = -\sum_{j=1}^{N} \frac{\partial \phi_i^g}{\partial z_{ij}} \cdot k_j = -(\nabla_z \phi_i^g)^T \boldsymbol{K} \tag{45}$$

where $\boldsymbol{K}$ is the key matrix (as defined in (11)) and $\nabla_z \phi_i^g = \left[\frac{\partial \phi_i^g}{\partial z_{i1}}, \ldots, \frac{\partial \phi_i^g}{\partial z_{iN}}\right]^T$ is the attention weight vector. In (5), we defined $\phi_i^g$'s to be the **log-sum-exp** function and that led the attention weight vector $\nabla_z \phi_i^g$ to be the Softmax function. Now in the general case, if $\phi_i^g$'s are continuous and strictly convex, we can write (See [15] Proposition 1.3):

$$\nabla_z \phi_i^g(z) = \arg \max_{p \in \text{dom}(\phi_i^{g*}) \subset \mathbb{R}^N} \left[ p^T z - \phi_i^{g*}(p) \right] \tag{46}$$

where $\text{dom}(f)$ is the domain of function $f(\cdot)$, and $\phi_i^{g*}(p) = \sup_{z \in \text{dom}(\phi_i^g)} \left[ p^T z - \phi_i^g(z) \right]$ is the convex conjugate of $\phi_i^g(z)$. For the **log-sum-exp** function $\phi_i^g(z) = \log \left[ \sum_{j=1}^{N} \exp(z_{ij}) \right]$, the convex conjugate is the negative Shannon Entropy $\phi_i^{g*}(p) = \sum_{j=1}^{N} p_{ij} \log p_{ij}$.

On the other hand, in sparse attention [22], the attention weight vector $\nabla_z \phi_i^g(z)$ is set to be the $\alpha$-entmax function which has the exact same form as (46) with $\phi_i^{g*}(p) = -\mathcal{H}_\alpha^T(p)$, where

$$\mathcal{H}_\alpha^T(p) = \begin{cases} \frac{1}{\alpha(\alpha-1)} \sum_{j=1}^{N} \left( p_{ij} - p_{ij}^\alpha \right), & \alpha \neq 1 \\ -\sum_{j=1}^{N} p_{ij} \log p_{ij}, & \alpha = 1 \end{cases} \tag{47}$$

is the Tsallis continuous family of entropies [89]. It is straightforward to show that for $\alpha = 1$ (*i.e.* the Shannon Entropy), the $\alpha$-entmax function reduces to the Softmax function. However, as we saw before, we can alternatively derive the Softmax function by first deriving the energy component $\phi_i^{g*}(p)$ as the **log-sum-exp** function and then computing its gradient. Now by following the same process for the general Tsallis entropy, we can derive the equivalent energy component whose gradient would be the $\alpha$-entmax function. In particular, by setting $\phi_i^{g*}(p) = -\mathcal{H}_\alpha^T(p)$ (as done in the formulation of Sparse attention [22]), we will have the energy component $\phi_i^g(z) = [-\mathcal{H}_\alpha^T(p)]^*$, which can be further derived in closed form as:

$$\phi_i^g(z) = \frac{1}{\alpha(\alpha-1)} + \sum_{j=1}^{N} y_{ij}^{1/(\alpha-1)} \left( z_{ij} - \frac{1}{\alpha(\alpha-1)} y_{ij} \right) \tag{48}$$

where
$$y_{ij} = \mathrm{ReLU}[(\alpha - 1)z_{ij} - \tau_i] \tag{49}$$

and $\tau_i$ is the Lagrange multiplier corresponding to the $\sum_{j=1}^{N} p_{ij} = 1$ constraint. Note that, in general, $\tau_i$ is a function of all $z_{ij}$'s; that is, $\tau_i = \tau(z_{i0}, \ldots, z_{iN})$. By plugging (48) into (44), we arrive at the equivalent energy function for the general $\alpha$-entmax attention (*i.e.* the sparse attention):

$$\phi^g(Q, K) = -\frac{1}{N} \sum_{i=1}^{N} \sum_{j=1}^{N} y_{ij}^{1/(\alpha-1)} \left( z_{ij} - \frac{1}{\alpha(\alpha-1)} y_{ij} \right) + C, \text{ where } z_{ij} = q_i^T k_j \tag{50}$$

## K.2 The Hierarchical Generalization

Now that we have the energy function for sparse attention in the flat case ((50)), we can generalize it to the hierarchical structure of nested signal by following similar recipe as (8). In particular, for node $A$ in the signal hierarchy $h_x$, the *hierarchical sparse energy* is recursively defined as:

$$\phi_\alpha(A) = - \sum_{B \in chd(A)} \frac{|\ell(B)|}{|\ell(A)|} \phi_B^g \left( -\phi_\alpha(B), \log |\ell(C_1)| - \psi_{B \to C_1}, \ldots, \log |\ell(C_k)| - \psi_{B \to C_k} \right) \tag{51}$$

where $C_1, \ldots, C_k$ are the sibling nodes of $B$, $\psi_{B \to C_k}$ is the interaction energy function defined in (7), and the multi-variate function $\phi_B^g$ has the same functional form as (48). Then (8) can be seen as a special case of where $\alpha = 1$ and $\phi_B^g$ reduces to the **log-sum-exp** function. Given the hierarchical sparse energy, we can derive the hierarchical sparse attention by taking the gradient of $\phi_\alpha(R_x)$ w.r.t to each query vector $q_i$, similar to the derivation in (9) for the Softmax case. We leave further derivation of an efficient algorithm and theoretical optimality for sparse attention to future work.

Lastly, it should be noted that similar to flat sparse attention, one can also learn the sparsity factor $\alpha$ via back-propagation in the hierarchical case. This can be further extended to learning different sparsity patterns for different levels of hierarchy, which can be useful depending on the application.

# L   Zero-shot Approximation of Self-attention: Ablation Study

In this appendix, we further expand on the experimental results for the zero-shot HSA approximation of RoBERTa presented in the main paper. In particular, we study the effects of approximating different combination of layers as well as different hierarchical structures for the datasets reported in the main paper.

## L.1   Experimental Setup

**Datasets**: We have run experiments on 5 GLUE datasets (SST-2, CoLA, MRPC, RTE and QNLI) as well as the AGNEWS and IMDB datasets.

**Models**: For each dataset, we have used the appropriate pre-trained RoBERTa checkpoint and configuration that has been fine-tuned on the corresponding task. Table 9 lists the checkpoints and configuration used for each dataset. All of our experiments involve only evaluation of pre-trained RobERTa without any training of fine-tuning it.

**Metrics**: We have computed Accuracy, Precision, Recall and F1 Score to measure the accuracy drop of pre-trained RoBERTa as its various layers are approximated by HSA.

**Impacted Layers**: As mentioned in the main paper, approximating all self-attention layers of RoBERTa typically leads to significant zero-shot accuracy drop across all tasks. However, approximating a subset of layers can introduce more reasonable gap while still benefiting from HSA speed up in terms of the number of FLOPs. Nevertheless, finding the best layer combination is a combinatorial problem. To alleviate this issue, instead of examining all different combinations, we only look at certain combinations based on two empirical observations. In particular, we observed that earlier layers in the network are typically more sensitive to approximation, whereas the latter ones are more amenable to it. This observation intuitively makes sense because the sooner approximation takes place in the network, the higher approximation error accumulates along the network. Moreover, having consecutive layers approximated typically increases the accuracy gap whereas interleaving them with regular self-attention layers decreases the gap.

| Dataset | RoBERTa Configuration | Checkpoint |
|---------|----------------------|------------|
| IMDB | Large | siebert/sentiment-roberta-large-english [a] |
| AGNEWS | Base | cardiffnlp/twitter-roberta-base-sentiment [b] |
| SST-2 | Base | textattack/roberta-base-SST-2 [c] |
| CoLA | Base | textattack/roberta-base-CoLA [d] |
| MRPC | Base | textattack/roberta-base-MRPC [e] |
| QNLI | Base | textattack/roberta-base-QNLI [g] |
| RTE | Base | textattack/roberta-base-RTE [h] |

Table 9: Checkpoints and RoBERTa configurations used for evaluating each task.

[a] https://huggingface.co/siebert/sentiment-roberta-large-english
[b] https://huggingface.co/cardiffnlp/twitter-roberta-base-sentiment
[c] https://huggingface.co/textattack/roberta-base-SST-2
[d] https://huggingface.co/textattack/roberta-base-CoLA
[e] https://huggingface.co/textattack/roberta-base-MRPC [f]
[g] https://huggingface.co/textattack/roberta-base-QNLI
[h] https://huggingface.co/textattack/roberta-base-RTE

Based on these two observations, in our experiments, we only examine combinations where a start layer (denoted by SL) and every other layer after that are approximated by HSA. The X-axis for the bar plots in this section is associated with SL. Also, the right bar in each plot represents the metrics for the original model without HSA approximation.

**Hierarchy**: For these experiments we chose to use fixed hierarchies based on non-overlapping hopping windows rather than semantic hierarchies based on the text structure. The reason behind this choice is that semantic hierarchies (such as sentences, paragraphs, etc.) are example dependant which means they would incur different number of FLOPs for different examples. But since our ultimate goal from this experiment is to reduce the number of flops consistently across the data, we opted to use fixed hierarchies.

The fixed hierarchies here are characterized by having a fixed branching factor for all the nodes belonging to the same level of the hierarchy. We then denote such hierarchy by the tuple $(A, B, C, ...)$ where $A$ is the branching factor at the lowest level of the hierarchy, $B$ is the branching factor for the next level and so on. Having this notation in place, we have experimented with the following hierarchies:

1. $(2, 2, 2, 2)$: A hierarchy with low branching factor at all levels.

2. $(2, 4, 8, 16)$: A hierarchy with low branching factor on the bottom and high branching factor on the top.

3. $(7, 7, 7, 7)$: A hierarchy with high branching factors at all levels.

4. $(8, 4, 2)$ A hierarchy with high branching factor on the bottom and low branching factor on the top.

## L.2 Results

**SST-2 Task**: As Figure 7 shows, the SST-2 task is relatively robust to the choice of SL (start layer for HSA approximation), where the accuracy gap widens if SL falls below Layer 5. Also, the choice of hierarchy is relatively inconsequential except for the narrow hierarchy with low-branching factor across all its levels, which demonstrates slightly poorer results compared to the rest.

**RTE Task**: As Figure 8 shows, the RTE task exhibits the same behavior as the SST-2 task with a major accuracy drop takes place when SL falls below Layer 7. Different hierarchy structures seem to have similar behavior though.

**MRPC Task**: As Figure 9 shows, for the MRPC task, the layers with even index seem to be way more sensitive to HSA approximation than the odd-index layers. Among odd index layers, the accuracy gap starts to widen for SL below Layer 7. As for hierarchy structures, the structures with low branching factor on the bottom levels seem to do better than the other two candidates.

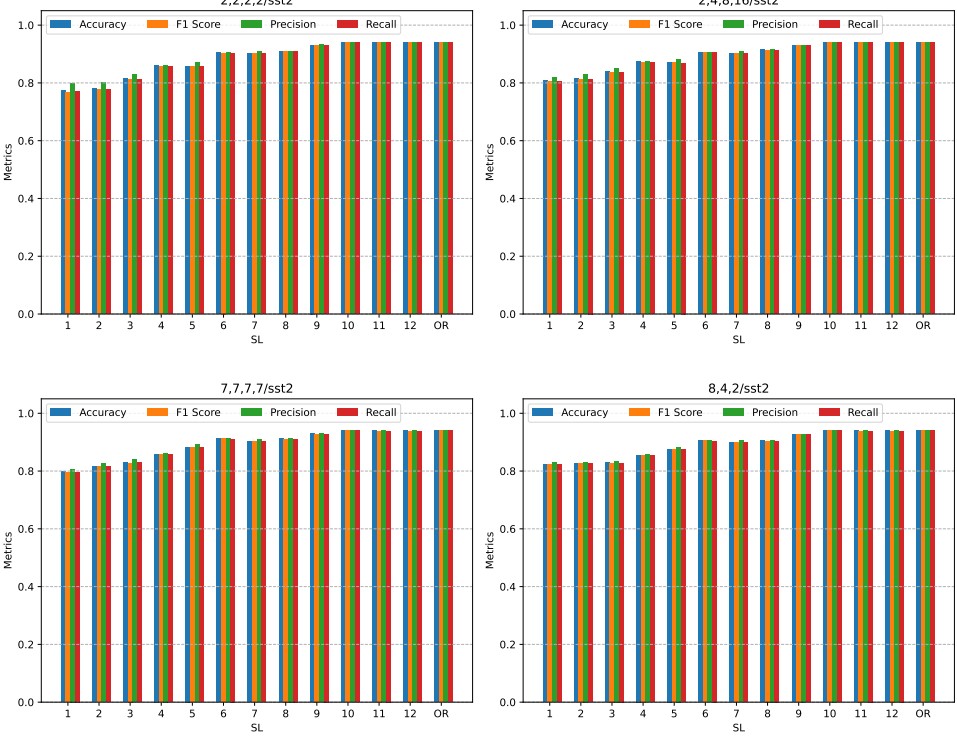

Figure 7: The accuracy metrics of HSA approximation of self-attention layers in RoBERTa for the STT-2 task.

**QNLI Task**: As Figure 10 shows, for the QNLI task, there is a sharp drop of accuracy when SL falls below Layer 8, whereas for the last four layers the accuracy drop is practically insignificant. This shows that in this case, the last 5 layers are quite amenable to approximation. As for the hierarchy structures, they do not exhibit any significant difference for this task.

**CoLA Task**: As Figure 11 shows, similar to the MRPC task, in CoLA task, the layers with even index seem to be way more sensitive to HSA approximation than the odd-index layers. However, unlike the MRPC task, the hierarchy structures with high branching factor on the bottom seem to significantly perform better than the ones with low branching factor on the bottom.

**AGNEWS Task**: As Figure 12 shows, for AGNEWS task, we can pretty much start SL at Layer 2 and as long as we approximate every other layer, the accuracy drop in insignificant. As for hierarchy structures, we have tested only 2 of our structures with this datasets, but did not observe any significant difference.

**IMDB Task**: Unlike the previous tasks, for IMDB task, we use RoBERTa-large with 24 layers. As Figure 13 shows, as long as SL stays above Layer 15, the accuracy drop is insignificant. Also some layers like Layers 10 and 15 seem to be moresensitive if we start the HSA approximation from them. As for hierarchical structure, among the two candidate we used for this task, the one with high branching factor on the bottom seems to do much better.

## M   Limitations

In this section, we state some of the main limitations of the current work both in terms of the proposed framework itself as well as our process of evaluating it in this paper.

As for the HSA framework itself, we would like to emphasize that it only operates on *tree-based*, compositional information hierarchies that are already *given* or *extractable* using a preprocessing procedure. In other words, HSA does *not* automatically learn the hierarchical structure embedded in

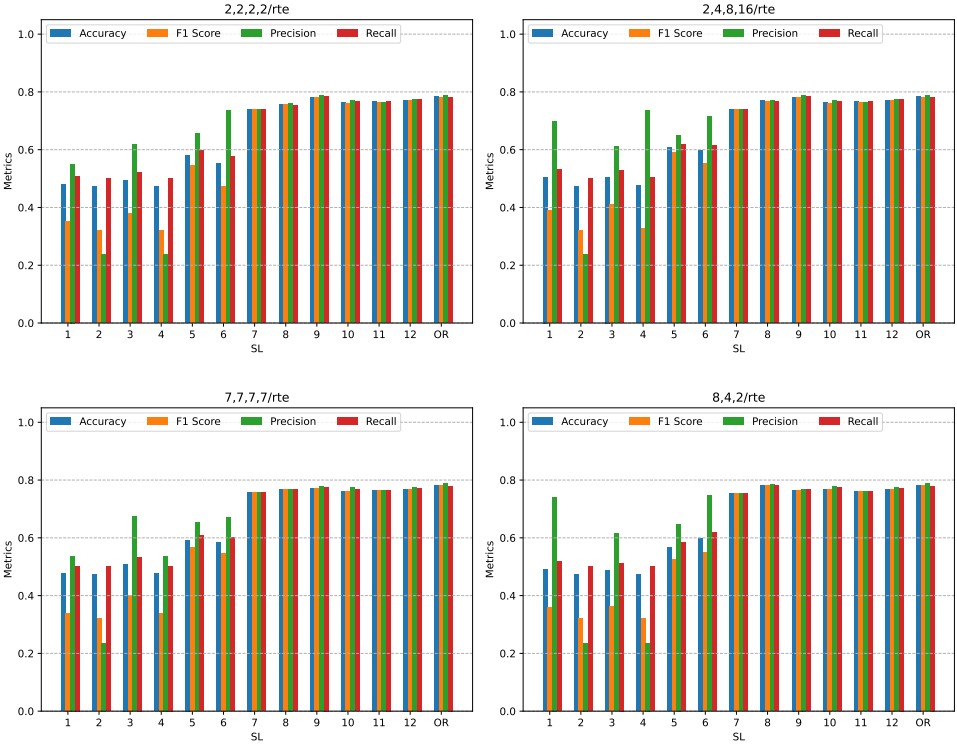

Figure 8: The accuracy metrics of HSA approximation of self-attention layers in RoBERTa for the RTE task.

the data. Furthermore, our proposed framework does *not* introduce any additional learnable parameter across the hierarchy on top of the standard self-attention parameters. While this is a useful feature in certain usecases such as zero-shot HSA replacement post-training as described in Section 5.3, in some other scenarios, this would introduce a limitation in terms of the learning capacity of our framework. This also makes it quite challenging to perform a fair comparison of our framework against other hierarchical attention mechanisms as in pretty much all of those frameworks, extra hierarchy-realted, learnable parameters have been incorporated as a fundamental component of the framework.

As for our evaluation process in this paper, we understand that the scope of our empirical study does not include all mainstream applications, especially in the auto-regressive generation domain. However, we would like to note that this work primarily focuses on the theoretical foundations of the proposed nested signal data structure and its hierarchical attention mechanism. We hope that our work can be used as the foundation for many follow-up efforts focusing on various applications with potential technical extensions. Nevertheless, we have still provided the initial theoretical extensions for some of the major potential follow-ups to the present work, such as hierarchical auto-regressive generation (Appendix G) and hierarchical sparse attention (Appendix K).

Finally, we would like to note that the goal of training foundational models using HSA to incorporate the inherent hierarchical inductive biases of our human knowledge base is indeed inspirational, as we do not have the computational resources required for that scale of training. However, from a theoretical point of view, our HSA method provides a plausible framework to achieve this goal in practice.

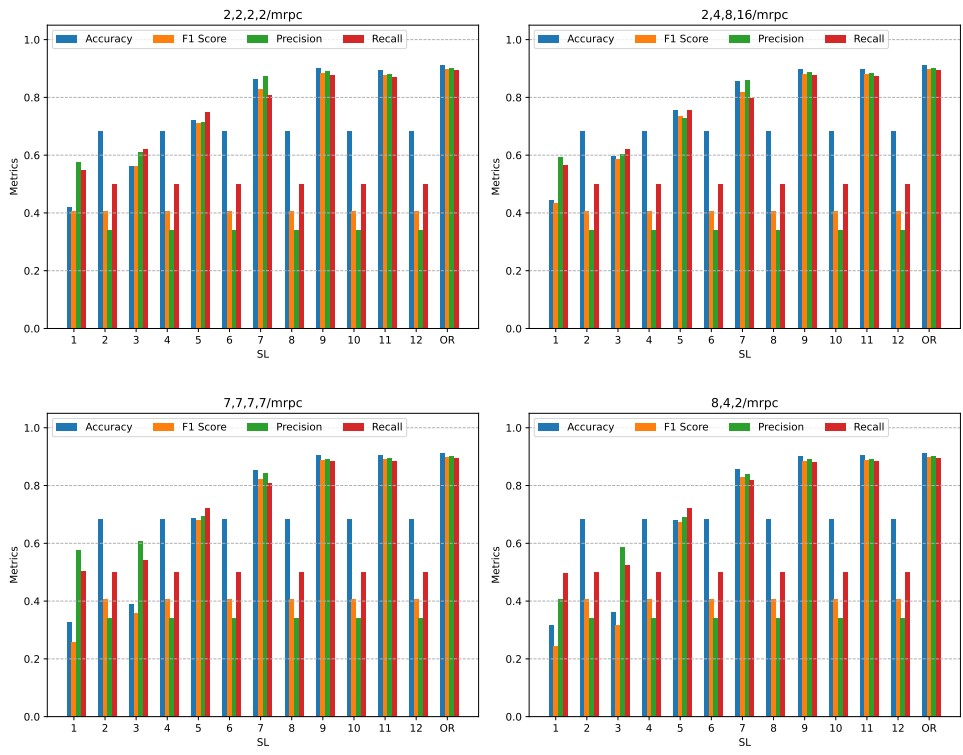

Figure 9: The accuracy metrics of HSA approximation of self-attention layers in RoBERTa for the MRPC task.

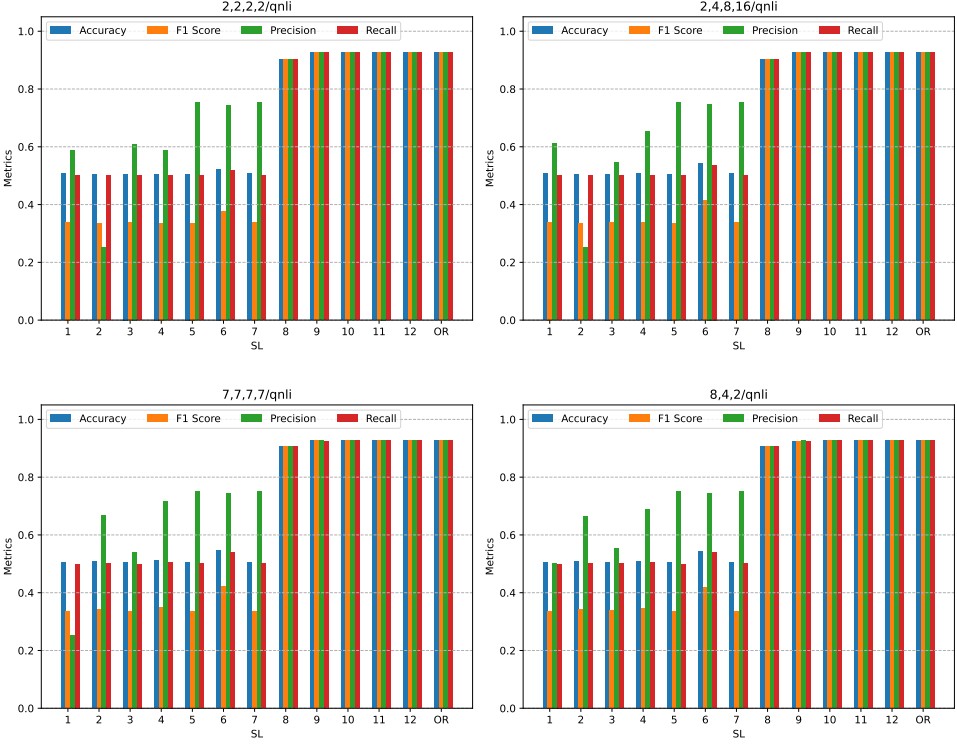

Figure 10: The accuracy metrics of HSA approximation of self-attention layers in RoBERTa for the QNLI task.

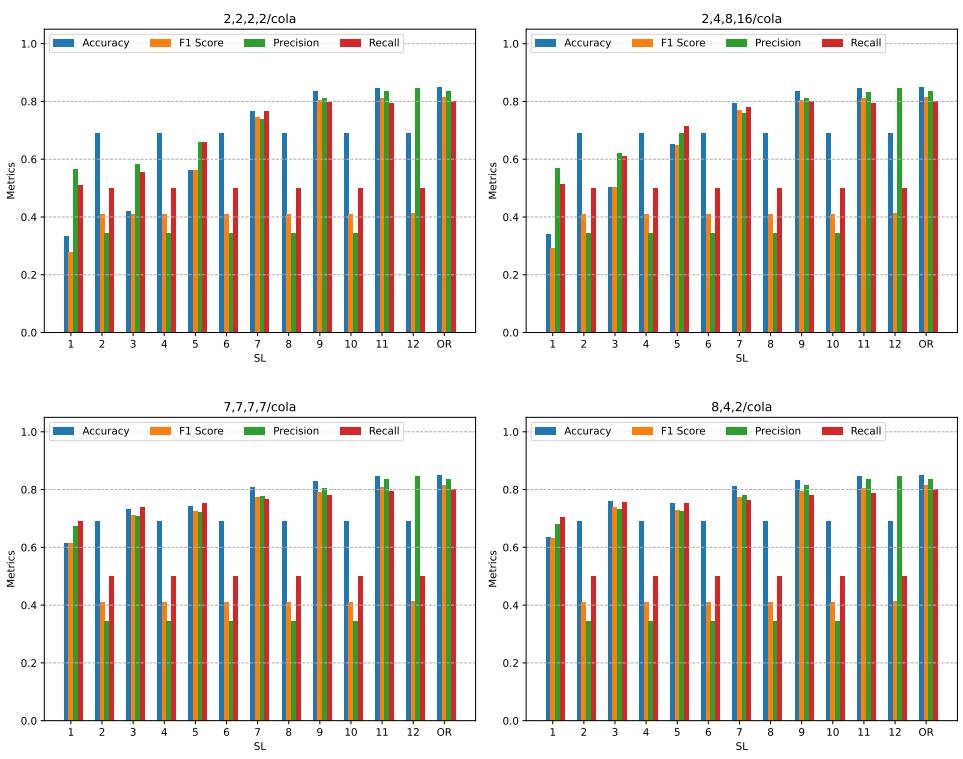

Figure 11: The accuracy metrics of HSA approximation of self-attention layers in RoBERTa for the CoLA task.

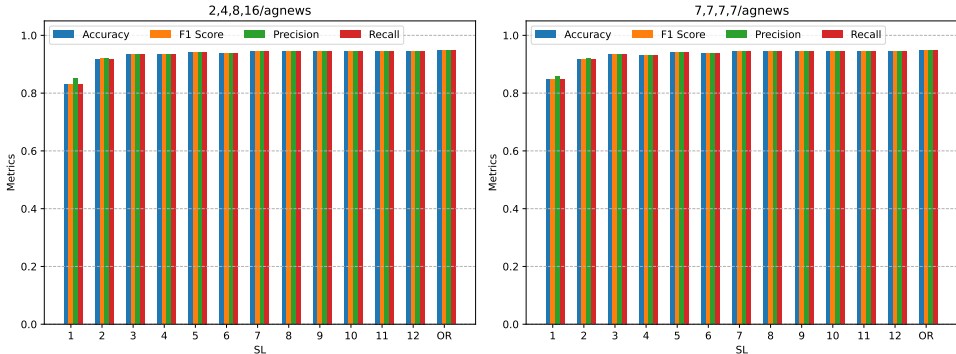

Figure 12: The accuracy metrics of HSA approximation of self-attention layers in RoBERTa for the AGNEWS task.

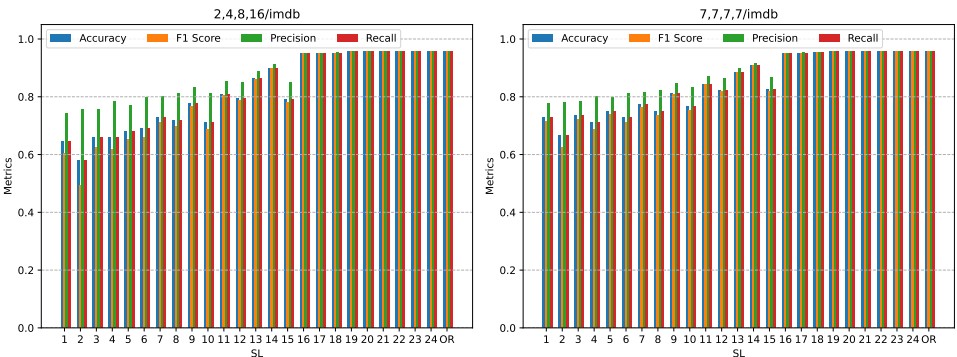

Figure 13: The accuracy metrics of HSA approximation of self-attention layers in RoBERTa for the IMDB task.

