# OpenReview forum: "Hierarchical Self-Attention: Generalizing Neural Attention Mechanics to Multi-Scale Problems"
_NeurIPS.cc/2025/Conference — NeurIPS 2025 poster_

### Official Review · Reviewer_tJ56 · 2025-06-30

**Clarity:** 2
**Significance:** 3
**Originality:** 4
**Rating:** 4
**Confidence:** 2

**Summary:**

The paper presents a new mechanism attention that inherently supports both multi-modal and multi-scale problems. It starts by introducing a formalism of multi-scale problems as trees of nested signals. It then uses this formalism and the following logic to obtain a new formulation of the attention mechanism. First, given key K and queries Q, one can express the conditional entropy of Q given K, and try to minimize it to increase the information. Second, because this entropy is intractable, one expresses an upper bound using variational distribution. By considering a Boltzman distribution and computing the gradient descent for the query, we can obtain new formulations of the attention mechanism which depend on the energy function of the Boltzman prediction. With a particular energy function we can even retrieve an attention quite similar to the original softmax attention. Using another energy function one can rewrite the attention matrix as a block matrix. It means, still considering the hierarchical structure of the problems as a tree, attention value is uniform across leaf elements that belong to different branches. This effectively reduces the number of values to learn.

The first experiment evaluates the performance of this new hierarchical attention on sentiment analysis. Texts are split into paragraphs, sentences and tokens. The hierarchical attention outperforms the traditional one. The second experiment evaluate news classification in a multi-modal context. The proposed attention yield to slightly worse results but for a much cheaper computation.

**Questions:**

- Could comparison with models dedicated to hierarchical settings like Swin Transformer be added?
- Would it be possible to reduce the section about finding back a similar attention as the traditional softmax one, to bring forward the per-block attention matrix? It would be also the opportunity to bring forward the dynamic programming method to compute the hierarchical attention, which is pointed as a major contribution but relegated to the appendix.

**Ethical Concerns:**

["NO or VERY MINOR ethics concerns only"]

**Final Justification:**

The paper presents an interesting hierarchical attention mechanism, which promises to leverage relational organization of the data at a lower computational cost than traditional self attention. The paper already good in its current form, would greatly benefit from some reorganization to convey a clearer message. For instance, it might be more important to expose the dynamic programming approach to reduce complexity than the ability of the proposed formalism to recover the traditional softmax attention. Additionally, the experimental settings could be strengthened to illustrate further the benefits of the proposed method. This could include use of the new attention mechanism in traditional model architectures on diverse tasks.
Finally, I believe the idea is interesting enough for the paper to be accepted. Other reviewers made significant remarks that could be integrated in the final version.

**Limitations:**

More experimental evaluation of the hierarchical self-attention would help assessing its limitations.

**Quality:**

3

**Strengths And Weaknesses:**

## Strengths

- The paper presents an original per-block attention that natively  support for multi-modal and hierarchical settings
- Experimental results show the new attention mechanism performs on par with the traditional one for a cheaper cost.

## Weaknesses

- The paper is dense, which makes it difficult to follow despite of the clarity of the different exposition steps. The paper may gain in readability with some reorganisation. Some parts of the mathematical justification behind the new attention mechanism may be superfluous. For instance, is it necessary to emphasize so much how to find back with the new formulation an attention that is close to the traditional one? The per-block attention matrix that supports hierarchical settings should be brought forward sooner in the paper.

- This new attention formalism has the potential to impact many problems due to its support for hierarchical and multi-modal settings. The experimental settings could be larger to assess this impact. For instance, it could include comparison with other methods that were design with hierarchy in mind like Swin Transformer.

---

> ### Author Rebuttal · Authors · 2025-07-30
>
> We sincerely thank the reviewer for their insightful comments on our submission.
>
> 1. **As for the organization of the paper,** we agree with the reviewer that our manuscript is somewhat dense as we have tried to cover many different bases and angles, leading to an additional 25 pages of appendix materials. In order to make the final manuscript more clear, we will move the notion of block attention higher to the beginning of the paper as the reviewer suggested. If possible, we will also try to move some of the background materials to the appendix to find more room for the Dynamic Programming section in the main text. As for the section on the KL-optimality of HSA (i.e. Theorem 3.2), we would like to note that this result is a crucial theoretical contribution of our work as it establishes the optimality of HSA approximation among other block approximations of the attention matrix. But why is this result important? Because it opens the door for another equally important application of HSA in practice: the zero-shot plug-n-play capability of HSA in integrating into the pretrained models, as it was empirically validated in Section 4.3 and Appendices J and L. Such plug-n-play capability effectively enables us to avoid building large-scale HSA models from scratch but from already pre-trained transformers. Our theoretical result guarantees the optimality of such zero-shot procedure.
>
>
> 2. **As for comparison to some other hierarchy-based models such as Swin-Transformer,** we are planning to add such baselines to some of our experiments. However, making a fair comparison of HSA to these frameworks is somewhat tricky. In particular, most of such models including Swin-Transformer encode a fixed hierarchical structure within their architecture by assigning different sets of $Q$, $K$ and $V$ projection parameters to different levels of the hierarchy. While this restricts these models' capability in handling arbitrarily-shaped hierarchies (unlike HSA), it also introduces a completely different parameter structure than that of the HSA. In other words, any potential gain or loss cannot be properly attributed to the hierarchical attention mechanism only, which is the main focus of the current work. Please note that in all of our current experiments, the baselines as well as their rival HSA model have exactly the same parameter structure and count but different attention calculation procedures, which makes the comparison fair and the results meaningful. Nonetheless, we are currently trying to come up with a variation of some of these hierarchical baselines that have more similar parameter structure to HSA, but that would require training such baselines from scratch.

---

> > ### Comment · Reviewer_tJ56 · 2025-08-05
> > **Answer**
> >
> > I thank the authors for answering my questions.
> >
> > I maintain my evaluation as I believe the paper organization could be improved for the final draft.
> > Regarding baseline with other hierarchical methods, even fixed ones, I believe there is still an interest in comparing performances, running times and resources required.

---

### Official Review · Reviewer_HMnn · 2025-06-30

**Clarity:** 3
**Significance:** 3
**Originality:** 3
**Rating:** 5
**Confidence:** 4

**Summary:**

The paper introduced hierarchical structure into the soft-max attention mechanism in a mathematically interpretable manner. The authors first derived the softmax attention layer as similar to one step gradient update of an entropy minimization problem. Then generalize the attention layer to nested signals by introduce hierarchical structure into the variables/nodes

**Questions:**

NA

**Ethical Concerns:**

["NO or VERY MINOR ethics concerns only"]

**Final Justification:**

I appreciate the response/rebuttal from the authors. I will maintain my evaluation.

**Limitations:**

+ Have the authors try to do more than 1 gradient step update ?
+ How the performance vary according to different signal hierarchy ? Is the a measurement to quantify the robustness of one hierarchy to another aside from doing the training process.

**Quality:**

3

**Strengths And Weaknesses:**

Strengths:
+ The network architecture are derived from unrolling, which leads to an interpretable network.
+ Introduce the hierarchical structure into the attention map.
+ Clear performance gain and also computations cost reduced due to hierarchical structure

Weaknesses:
+ Lack experiments to demonstrate the interpretability of the model, Ex: do more than 1 gradient steps ?, plot reduced entropy objective for various number step/layers ?
+ The signal hierarchy are not always obvious for various datasets/application: graphs signals. The paper don't introduce any mathematical methods to create hierarchy from data and mainly use intuition.

---

> ### Author Rebuttal · Authors · 2025-07-30
>
> We truly appreciate the reviewer's constructive feedback and comments.
>
> 1. **Regarding lack of experimentation for multi-step gradient-descent interpretation,** we'd like to note that the main angle and focus of this work is the theoretical derivation of hierarchical attention from the entropy minimization interpretation of self-attention. In that sense, the gradient-descent interpretation is a secondary byproduct of our theoretical framework which is, of course, worth exploring on its own in a separate work. At the same time, we'd like to mention that performing $n$-step gradient descent on conditional entropy using the attention operator is mathematically similar to having $n$ sequential self-attention layers that share weights. But weight sharing in transformers has been already studied in the literature and shown to improve the performance (e.g. see [A, B, C]).
>
> [A] Xiao, Tong, et al. "Sharing attention weights for fast transformer." arXiv preprint arXiv:1906.11024 (2019).
>
> [B] Reid, Machel, Edison Marrese-Taylor, and Yutaka Matsuo. "Subformer: Exploring weight sharing for parameter efficiency in generative transformers." arXiv preprint arXiv:2101.00234 (2021).
>
> [C] Kowsher, Md, et al. "Does Self-Attention Need Separate Weights in Transformers?." arXiv preprint arXiv:2412.00359 (2024).
>
> 2. **Re: "a mathematical approach for creating the hierarchy",** as we stated in the paper, the HSA framework by itself doesn't provide any systematic approach to infer the hierarchy from data. Instead, the hierarchical structure is seen as domain knowledge which can act as inductive bias and subsequently encoded via the HSA framework. This leaves the door open for HSA to integrate with any manual or machine-learned hierarchy extraction method. Nevertheless, as shown by the results in Section 4.3 and Appendices J and L, in the absence of the hierarchical knowledge, one may still impose an "artificial" hierarchy on the input signal and still benefit from the regularization and speedup characteristics of HSA. Also, to further understand the sensitivity of HSA to the choice of the hierarchical structure (as the reviewer suggested), we have conducted ablation studies in Appendix L using four different imposed hierarchies across multiple datasets (please see Figures 7-13 in the appendix). As these experiments show, certain tasks can be more sensitive to the choice of the imposed hierarchy, while others are more robust. This further shows the strong correlation between the data geometry and the nature of the classification task and highlights the importance of cross-validation for picking a suitable hierarchical structure when prior domain knowledge is limited.

---

> > ### Comment · Reviewer_HMnn · 2025-08-05
> >
> > I appreciate the response/rebuttal from the authors. I maintain my evaluation.

---

### Official Review · Reviewer_NPM6 · 2025-07-03

**Clarity:** 2
**Significance:** 4
**Originality:** 3
**Rating:** 4
**Confidence:** 2

**Summary:**

This paper introduces a novel mathematical concept termed nested signal to formally represent hierarchical and multi-modal information within the Transformer framework. Based on this, this paper derives Hierarchical Self-Attention (HSA), a generalization of the standard Softmax attention mechanism that incorporates hierarchical structure. The paper further presents an efficient dynamic programming algorithm to compute HSA and demonstrates its effectiveness on both unimodal and multimodal tasks. Empirical results on IMDB, Elec, and N24News datasets show performance improvements over baselines.

**Questions:**

Is it possible to dynamically infer or adapt the hierarchy depth rather than requiring it to be pre-specified? How would the method generalize to scenarios with unknown or evolving hierarchical structures?

Can the authors provide further empirical evidence for the multi-modal applicability of the proposed method, ideally on more diverse and competitive benchmarks?

How does HSA interact with modern large-scale pre-trained vision-language models or other Transformer variants? Could HSA be integrated into such models, and if so, what would be the expected benefit?

**Ethical Concerns:**

["NO or VERY MINOR ethics concerns only"]

**Final Justification:**

I have read the rebuttal and the comments from other reviewers. While the rebuttal partially addresses my concerns, I believe there is still room for improvement—particularly in the experimental evaluation on multi-modal tasks. Based on the overall merits and the clarifications provided, I am updating my score to Weak Accept.

**Limitations:**

The requirement to pre-define the hierarchy limits the method's flexibility and may restrict its applicability to tasks with fixed or clearly known hierarchical structures.

The experimental evaluation for multi-modal scenarios is limited, relying on a single dataset with relatively weak baselines, making it hard to draw strong conclusions about multi-modal performance.

The computational overhead of the proposed HSA, despite claimed efficiency improvements, is not thoroughly analyzed or compared to standard attention in practical large-scale settings.

**Paper Formatting Concerns:**

Overall, the paper is well-organized and clearly written.

**Quality:**

3

**Strengths And Weaknesses:**

# Strengths:

The nested signal concept is new and interesting.

The design of Hierarchical Self-Attention appears principled and technically sound.

Addressing multi-level, multi-modal learning within a unified Transformer-based framework is an important direction, especially given the increasing complexity of modern AI tasks.

The proposed method shows empirical improvements on several benchmarks (IMDB, Elec, N24News), indicating its potential practical utility.

# Weaknesses:

Based on my understanding, the proposed method requires the hierarchy depth to be explicitly pre-defined. This rigid design may limit its flexibility and scalability, especially when dealing with tasks or datasets where the hierarchical structure varies or is not clearly defined. This contrasts with traditional Transformers, which often benefit from scaling data or model size without making strong structural assumptions.

Although the paper claims to address multi-modal scenarios, only the N24News dataset is used to demonstrate this aspect. Moreover, the reported improvements over baselines like FSA are relatively modest, and the chosen baselines are not particularly strong or recent.

While I agree with the authors' experimental design principle of isolating contributing factors, limiting multimodal evaluation to a single dataset with weak baselines undermines the strength of the multi-modal claim. Additional, more convincing evidence is needed to support this aspect.

---

> ### Author Rebuttal · Authors · 2025-07-30
>
> We are sincerely grateful to the reviewer for their insightful feedback and bringing up some important points that'd require our further explanation.
>
> 1. **Re: "predefined depths",** we would like to emphasize that our proposed HSA framework does NOT require the input hierarchies to have any pre-defined depth, size or shape. In fact, within a minibatch, each nested signal example can have its own different depth, size and shape, and this is already supported via an efficient implementation using the *breath-wise tree concatenation* technique in our codebase, as explained in Appendix E.4. In other words, the same way that the classical transformer architecture is invariant to the length of the input (flat) sequence, the HSA-based transformer architecture is invariant to the depth, size and shape of the input nested signal. In that sense, HSA-based transformer can very well handle "evolving" hierarchies. Now, if one wants to define a separate $Q$, $K$ and $V$ projection for each different modality in each layer of transformer (as proposed in Appendix F), these modalities need to be predefined, so one can know how many of such projections are needed in each layer. However, once these modalities (e.g. language and vision) are known, one may compose arbitrarily-shaped hierarchies using these predefined modalities and feed them to the same HSA transformer model.
> That said, as we explicitly stated in the paper, the HSA framework cannot "infer" the hierarchy; instead, the hierarchy and its implicit geometry are seen as external domain knowledge acting as a form of inductive bias. In that sense, hierarchy inference is completely tangential to the HSA framework, enabling us to use HSA in combination with any hierarchy extraction method. It should also be noted that, depending on the scenario, even when the hierarchical structure of the input isn't given or learned, we may still benefit from HSA by imposing "artificial" hierarchies on the input signal, and effectively using it as a form of regularization. In fact, all the results reported in Section 4.3 and Appendices J and L are obtained by incorporating imposed hierarchies.
>
> 2. **Re: the multi-modal empirical study,** we understand that the reported results are somewhat limited to one task. However, we would like to note that most interesting multi-modal tasks such as VQA, visual-language reasoning, etc. almost always require auto-regressive generation (ARG) as a key component of the model. But in this work, we have deliberately tried to stay away from ARG because incorporating HSA with ARG capabilities is not trivial and merits its own dedicated, follow-up work. Nonetheless, we have laid the foundation for hierarchical ARG in Appendix G to be elaborated upon in future work. In the meantime, for the purpose of this work, we will make this point clear in the final version of the draft to avoid any empirically unsubstantiated claim or confusion.
>
> 3. **Re: integrating with large-scale vision-language transformers:** The HSA framework can be in principle integrated with any multi-modal transformer architecture including large-scale vision-language transformers. Currently, many of such models naively put signals from various modalities on a simple, flat sequential geometry and instead rely on massive-scale pretraining to learn the underlying geometry of cross-modal interactions from data. However, if such geometrical information is partially available as domain knowledge, HSA can incorporate it within the proposed nested signal data structure to potentially reduce the statistical complexity of pretraining and finetuning large-scale transformers. For instance, assume that the input data consist of images with their corresponding scene graphs (where the scene graph's nodes are detected object bounding boxes with some textual descriptions while its edges model various relationships between the objects). Feeding such rich data structure to classical vision-language transforms often involves losing most of such rich geometrical and semantical information in the process of "flattening" the signal, and instead relying upon massive-scale pretraining to recover part of that knowledge. Whereas, it's quite seamless to encode such prior knowledge using the nested signal construct and perform self-attention operation on it using the HSA formalism.
>
> 4. **Re: the computational overhead of HSA,** first please note that the main overhead of HSA is building the hierarchy for the input signal. But that can be seen as part of the tokenization process, and similar to tokenization, it can be done only once on the CPU before training. As for the rest of computational analysis, we have provided GFLOP metric on various tasks with different context lengths in Table 3. As shown in the table, HSA has significantly decreased the required GFLOPs compared to the baseline. Furthermore, the reduction is way more dramatic for larger context lengths, as shown in Table 3. As for the exact time comparisons of HSA and the baselines, we'd like to note that such comparisons are heavily implementation dependent and rely on various CUDA optimization tricks. However, efficient CUDA implementation of HSA is beyond the scope of our current paper, which is primarily meant to lay the theoretical foundations of the proposed framework.

---

### Official Review · Reviewer_edS5 · 2025-07-08

**Clarity:** 3
**Significance:** 2
**Originality:** 2
**Rating:** 4
**Confidence:** 2

**Summary:**

The paper introduces hierarchical self attention (HSA) to reason over data with hierarchies or multiple modalities. The paper introduces nested signal model for such data, and then perform entropy minimization to obtain HSA. They demonstrate that such a representation is KL-closest block-constrained approximation of standard soft max attention. The authors show that HSA increases accuracy on long-text sentiment tasks (ex: IMDB) and outperforms baselines such as DeepSet and vanilla transformer on multimodal news classification benchmark.

**Questions:**

Please see weakness section

**Ethical Concerns:**

["NO or VERY MINOR ethics concerns only"]

**Final Justification:**

Although most of my asks were in the negative in the rebuttal, the paper assessment still stands. I would keep my initial rating. Hopefully the authors have ideas for improving and extending the work.

**Quality:**

2

**Strengths And Weaknesses:**

Strengths
1. Main strength of the paper it's clean derivation from first principles with strong grounding provided by KL-optimality proof.
2. The dynamic programming implementation wit black box sub attention for plugging in fast approximation helps bring down the computation complexity significantly.
3. It can be used as dtop in replacement for softmax attention without increasing memory footprint.

Weaknesses
1. Main supervised tasks are on a limited scale (ex: classification), and there is no token level evaluation of generation limiting the ability to understand the qualitative improvements obtained for generalized inference.
2. The baselines such as DeepSet and flat attention for multimodal tasks are sparse, how does it perform on more complex tasks such as VQA, Image Description etc. where the model inference and training becomes complex.
3. Main limiting factor of the proposed approach seem to be assumption of a fixed hierarchy. Is it possible to learn the hierarchy or how can this method be scaled where no existing parser is available.

---

> ### Author Rebuttal · Authors · 2025-07-30
>
> We appreciate the reviewer's insightful feedback and recognition of our fundamental derivation of hierarchical attention.
>
> 1. **Re: the scope of our empirical study,** we agree with the reviewer that our empirical study is somewhat limited to classification tasks and has not covered generative tasks which are key to modern LLMs. However, as we also stated in the main paper, addressing token generation is beyond the scope of the current work which is already somewhat dense presenting the theoretical foundations of the HSA framework. Nevertheless, in Appendix G, we have laid the theoretical foundations for hierarchical generation by proposing techniques to perform efficient causal masking and key-value caching under the hierarchical setting (which are both crucial to any realistic transformer-based LLM). Further empirical exploration of hierarchical generation has been deferred to future work.
>
> 2. **As for VQA and Image Description tasks**, we agree that both of these tasks involve more complex modality cross-interactions; however, those tasks are generative and, as mentioned above, outside the scope of our empirical study for this work, which is mainly focused on introducing the theoretical foundations for HSA and nested signals. Our goal is to address generative tasks in a follow up work dedicated to hierarchical generation.
>
>
> 3. **Re: learning the hierarchy,** as we have stated in our Limitations section, our HSA framework does not provide any mechanism to *learn* the hierarchal structure from data. In that sense, hierarchy is seen as a form of *domain knowledge* which injects strong inductive biases into the model and potentially reduces the statistical complexity of learning on top of improving the computational complexity. Nevertheless, depending on the scenario, one may be able to *impose* an artificial hierarchy in the absence of such domain knowledge or a parser. In fact, in Section 4.3 and Appendices J and L, we have *not* used any parser or domain knowledge for any of the experiments and all the reported results are based on purely imposed hierarchies (using various (fixed) branching factors at each level). As the results show, even with blindly imposed hierarchies for these problems, the HSA framework outperforms the baselines in terms of both accuracy and speed, highlighting the role of HSA as a form of regularization.

---

> > ### Comment · Reviewer_edS5 · 2025-08-05
> >
> > Thanks for the responses, although most of my asks are in the negative, but the paper assessment still stands. I would keep my initial rating. Hopefully the authors have ideas for improving and extending the work.

---

### Decision · Program_Chairs · 2025-09-17

**Decision:**

Accept (poster)

**Comment:**

Four reviewers recommend acceptance ratings. The main reasons for acceptance are the principled and original extension of attention to hierarchical and multi-modal data, the strong theoretical grounding, and the promising empirical results. The main weaknesses are the limited empirical scope, reliance on fixed hierarchies, and the need for broader evaluation and clearer presentation. The reviewers agree that the paper is technically solid and makes a valuable contribution, though there is room for improvement in future work.
On balance, the AC sees no basis to overturn the reviewer suggestions and recommends acceptance. The paper presents a theoretically grounded and practically promising approach to hierarchical self-attention, with clear potential for impact in multi-modal and hierarchical data scenarios. The AC highly recommends the authors to address the concerns of the reviewers and take into account their suggestions of improvement when preparing a revised version.